# Emergence of cortical network motifs for short-term memory during learning

Xin Wei Chia [1], Jian Kwang Tan [1,2], Lee Fang Ang [1,2], Tsukasa Kamigaki [1] & Hiroshi Makino [1] ✉

Learning of adaptive behaviors requires the refinement of coordinated activity across multiple brain regions. However, how neural communications develop during learning remains poorly understood. Here, using two-photon calcium imaging, we simultaneously recorded the activity of layer 2/3 excitatory neurons in eight regions of the mouse dorsal cortex during learning of a delayed-response task. Across learning, while global functional connectivity became sparser, there emerged a subnetwork comprising of neurons in the anterior lateral motor cortex (ALM) and posterior parietal cortex (PPC). Neurons in this subnetwork shared a similar choice code during action preparation and formed recurrent functional connectivity across learning. Suppression of PPC activity disrupted choice selectivity in ALM and impaired task performance. Recurrent neural networks reconstructed from ALM activity revealed that PPC-ALM interactions rendered choice-related attractor dynamics more stable. Thus, learning constructs cortical network motifs by recruiting specific inter-areal communication channels to promote efficient and robust sensorimotor transformation.

Brain-wide neural communications are refined during learning of adaptive behaviors[1-9]. Recent advances in simultaneous multi-regional single-cell recordings are expected to identify diverse operation principles beyond those discovered in single brain regions in isolation[10-16]. Probing brain-wide neural activity reveals a dynamic flow of information across regions, which may be subject to learning-dependent modulations[17-21]. Despite the technological progress, it remains underexplored how learning constructs cellular network motifs distributed in multiple brain regions.

Short-term memory is the ability to hold information online and considered critical for working memory, decision-making and motor planning. During short-term memory, neurons generate sustained activity in response to brief sensory inputs, bridging past and future events[22-28]. The persistent activity is maintained by recurrent positive feedback in local and brain-wide networks[29-34]. Recurrent functional connectivity generating the sustained neural activity may be reorganized during learning to facilitate more efficient and robust sensorimotor transformation.

The anterior lateral motor cortex (ALM) and posterior parietal cortex (PPC) in rodents have been extensively studied in isolation under two-alternative forced-choice tasks. Neurons in ALM show choice-selective activity during short-term memory for directional licking[35-40], while neurons in PPC are critical for perceptual discrimination, evidence accumulation and decision-making[41-49]. In delayed response tasks, choice-encoding attractor dynamics in ALM may be rendered more robust to internal and external perturbations by strengthening the recurrent connectivity[38,39]. Whether and how learning shapes multi-regional communications to achieve such robustness, however, are poorly understood.

We used calcium imaging with a two-photon random access mesoscope[11] to simultaneously record the activity of the same population of neurons from multiple cortical regions over the course of learning. Learning constructed functional network motifs for short-term memory where subnetworks consisting of neurons with similar task relevance were embedded in a sparsely connected global network. The cortical network motifs were further elaborated during learning by

[1]Lee Kong Chian School of Medicine, Nanyang Technological University, Singapore 308232, Singapore. [2]These authors contributed equally: Jian Kwang Tan, Lee Fang Ang. ✉e-mail: hmakino@ntu.edu.sg

selective strengthening of a region-specific communication channel between PPC and ALM. Our results provide evidence that learning augments efficiency and robustness for short-term memory via coordinated representation in distributed neural networks.

## Results

### Sensorimotor representation in the mouse dorsal cortex across learning

Head-restrained mice expressing the calcium indicator GCaMP6s[50] (CaMKII-tTA × TRE-GCaMP6s) were trained for 2-3 weeks in a delayed-response task. In this task, mice learned to localize a tactile stimulus presented either to their left or right whiskers (swept by brass rods at ~20 Hz, 1 s in duration) in randomly interleaved trials and respond by directional licking following a 2-s delay (Fig. 1a). At the end of the delay period, a go cue (green LED, 0.2 s in duration) signaled the beginning of a response window and both the left and right water spouts moved together and were made available to mice. The separation of the sensory instruction and the behavioral response in time during the

delay period required mice to engage short-term memory to generate appropriate actions. While correct responses were rewarded with sucrose water, incorrect responses were punished with a 0.5-s white noise and 8-s timeout. Task performance of individual mice improved gradually, reaching the mean correct rate of $85.0 \pm 9.6\%$ at the expert stage (naive stage: $22.9 \pm 9.5\%$; intermediate stage: $61.9 \pm 12.9\%$, mean $\pm$ SD, $P < 0.001$, $n = 7$ mice, one-way ANOVA, Fig. 1b).

To study how neural representation of task variables evolved across learning, we performed longitudinal two-photon calcium imaging of layer 2/3 excitatory neurons (~150-200 μm in depth) simultaneously from eight cortical regions of the left hemisphere (anterior lateral motor cortex, ALM; anterior and posterior regions of the primary motor cortex related to tongue and forelimb movement, respectively, M1a and M1p; secondary motor cortex, M2; primary somatosensory cortex for the forelimb and vibrissae, S1fl and vS1; retrosplenial cortex, RSC; posterior parietal cortex, PPC) with a two-photon random access mesoscope (2p-RAM)[11] (Fig. 1c). Responses and their learning-related changes were heterogenous across neurons

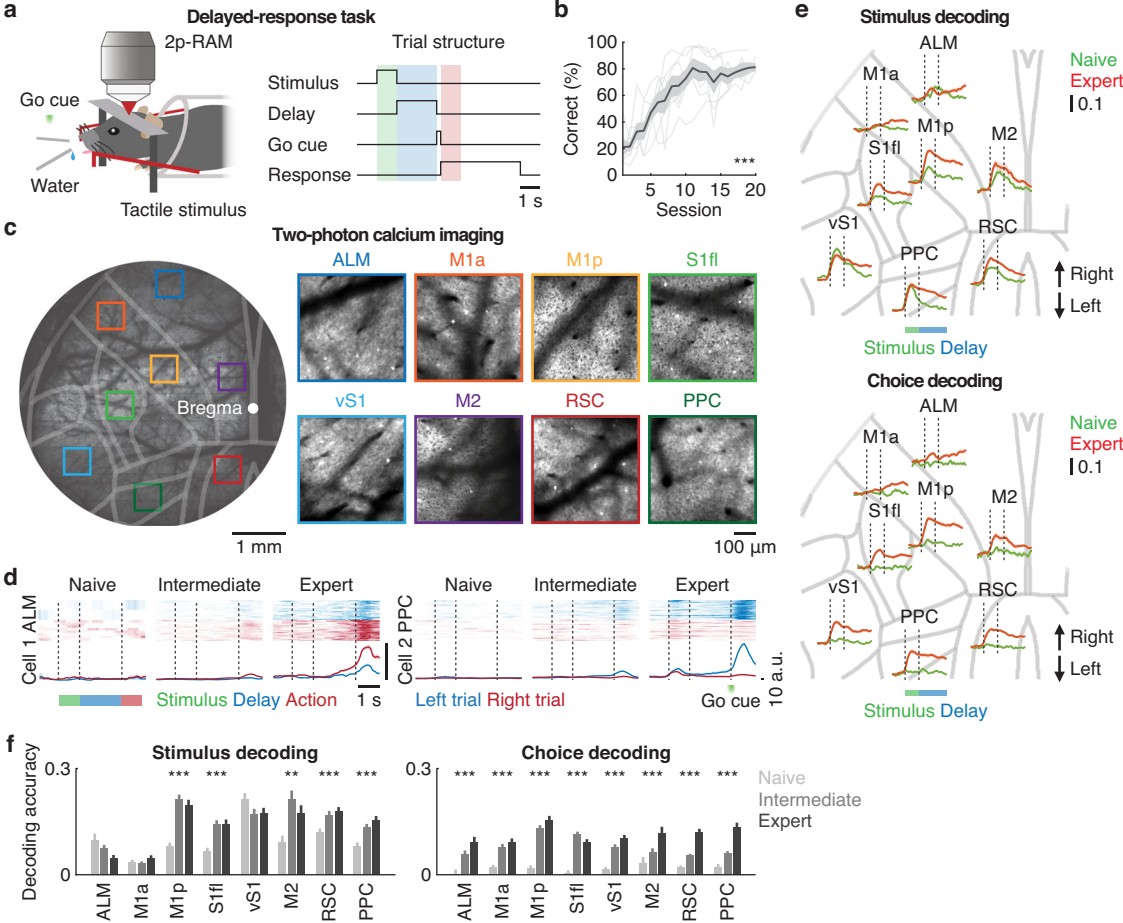

**Fig. 1 | Emergence of task representation in the mouse dorsal cortex across learning. a** Left. Schematic of the experiment. 2p-RAM: two-photon random access mesoscope. Right. Trial structure of the delayed-response task. **b** Learning curve for the delayed-response task (***$P < 0.001$, $n = 7$ mice, one-way ANOVA). Thick and thin lines indicate learning curves for mean and individual mice. Shaded area indicates mean $\pm$ SEM. **c** Example of two-photon calcium imaging in the mouse dorsal cortex. Single neurons from eight cortical regions were simultaneously imaged across learning (ALM: anterior lateral motor cortex; M1a and M1p: anterior and posterior regions of primary motor cortex; S1fl and vS1: primary somatosensory cortex for forelimb and vibrissae; M2: secondary motor cortex; RSC: retrosplenial cortex; PPC: posterior parietal cortex). Similar images were acquired across all mice. **d** Trial-by-trial (top) and mean (bottom) task-related activity of example

neurons in two trial types across the three stages of learning. Shaded areas indicate mean $\pm$ SEM. **e** Region-specific stimulus and choice decoding accuracy in single neurons over time prior to the go cue at the naive and expert stage. Shaded areas indicate mean $\pm$ SEM. **f** Quantification of region-specific stimulus and choice decoding accuracy in single neurons (***$P < 0.001$, **$P < 0.01$, stimulus: ALM: 40, 82, 82; M1a: 108, 115, 115; M1p: 122, 210, 210; S1fl: 111, 176, 176; vS1: 175, 196, 196; M2: 35, 82, 82; RSC: 184, 305, 305; PPC: 136, 184, 162 neurons from 7 mice; choice: ALM: 60, 82, 82; M1a: 108, 115, 115; M1p: 134, 210, 210; S1fl: 111, 176, 176; vS1: 175, 196, 196; M2: 48, 82, 82; RSC: 216, 305, 294; PPC: 166, 184, 162 neurons from 7 mice for naive, intermediate and expert, respectively, one-tailed bootstrap with false discovery rate, FDR). Error bars indicate mean $\pm$ SEM.

(Fig. 1d and Fig. S1), and we detected no epileptiform activity in the transgenic mice[51]. Area under the ROC (receiver operating characteristic) curve (AUC) analysis on single-cell activity showed the emergence of distinct information of stimulus and choice in each cortical region; consistent with previous studies[52,53], decoding accuracy for stimulus location increased in regions such as M1p, S1fl, M2, RSC and PPC, while decoding accuracy for choice increased globally across the entire dorsal cortex (one-tailed bootstrap for correlation with Benjamini-Hochberg false discovery rate, FDR, Fig. 1e and f).

To examine whether each cortical region maintained distinct information during the delayed-response task, we projected population activity in each cortical area onto axes that maximized stimulus and choice selectivity during the stimulus (1 s) and pre-action (0.5 s before the go cue) epoch, respectively[39]. Across learning, decoding accuracy for the stimulus location showed similar changes to the decoding accuracy derived from single neurons (one-tailed bootstrap for correlation with FDR, Fig. 2a), and the result remained essentially the same regardless of orthogonalization between the stimulus and choice axis. In contrast, choice selectivity increased globally across regions (one-tailed bootstrap for correlation with FDR, Fig. 2b), with ALM in expert mice displaying a pronounced increase in choice selectivity from the beginning of the delay period. We confirmed that the changes in the population choice activity were not due to changes in anticipatory licking because mice rarely showed licking behavior during the delay period when the water spouts were not accessible (less than 3% of all trials at the naive and expert stages).

## Emergence of coordinated activity for short-term memory across learning

ALM, a region of the frontal cortex important for lick-related choice representation during short-term memory, displays coordinated activity across hemispheres[54]. We hypothesized that similar coordination exists beyond single cortical regions and it is refined over the course of learning. We analyzed trial-by-trial correlations of population activity projected onto the stimulus and choice axes between pairs of cortical regions (Fig. 3a). The correlated stimulus and choice codes

across cortical regions were quantified by examining trajectory values along these axes during the stimulus and pre-action epoch, respectively. Along the stimulus axis, correlations between cortical regions remained relatively low within the same trial type across learning (one-tailed bootstrap, Fig. 3b and c). In contrast, correlations along the choice axis increased between regions such as ALM and PPC (one-tailed bootstrap, Fig. 3b and c). On the other hand, correlations between cortical regions along the stimulus and choice axis across both trial types generally became larger (one-tailed bootstrap, Fig. 3b and c). While trial-by-trial body movements became more similar across learning, the observed increase in activity correlations was not simply explained by the increase in movement stereotypy (Fig. S2a-d). We also ensured that the increased coordination was not sensitive to trial-type imbalance (Fig. S3a and b). These results demonstrate that there emerged distinct subnetworks for the stimulus and choice representation across learning.

## Cortical sparsening of functional connectivity across learning

We next studied how learning modifies cortical network motifs by restricting our analysis to the same population of neurons ($n = 1475$ from 7 mice) that were commonly identified across the three stages of learning (naive, intermediate and expert). To probe cortical network motifs, we built an encoding model (generalized linear model, GLM) for each neuron using the task variables (stimulus, delay, action, reward and forelimb movement) and directional functional coupling (preceding neural activity of other neurons) as predictors[55–57] (Fig. 4a and Fig. S4a). Encoding properties of task variables and directional functional couplings were statistically determined by removing each predictor and assessing the resulting model performance with pseudo-explained variance (E.V.)[58–60]. The full encoding model was further decomposed into the task-variable model and cell-coupling model by marginalizing the other predictors. The pseudo-E.V. of the cell-coupling model remained stable across learning (naive: $0.13 \pm 0.01$; intermediate: $0.13 \pm 0.01$; expert: $0.12 \pm 0.01$, mean $\pm$ SEM, $P = 0.36$, $n = 13$ sessions from 7 mice, repeated measures one-way ANOVA), indicating that the relative influence of other neurons' activities compared to that of task variables was persistently high (pseudo-E.V. of the

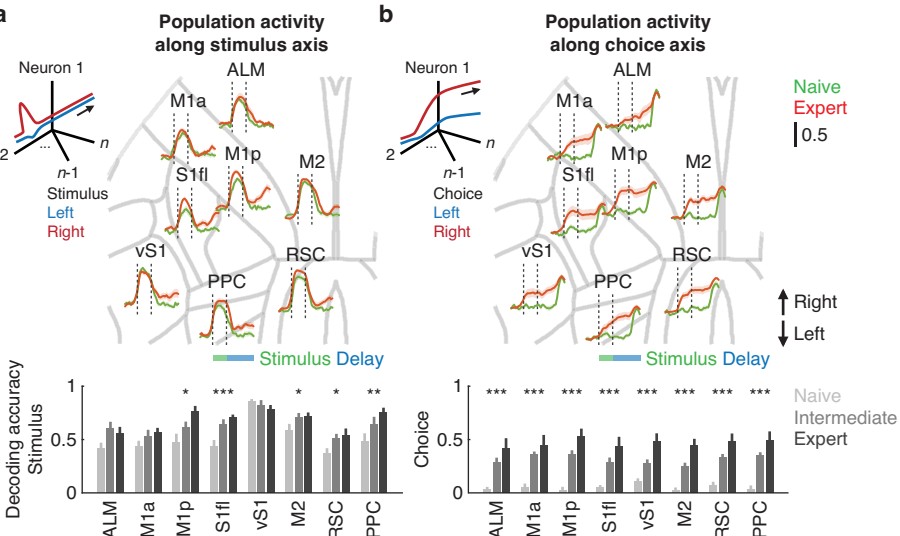

**Fig. 2 | Population activity projected to the stimulus and choice axis. a** Left. Schematic of dimensionality reduction to capture neural population activity along the stimulus axis. Right. Region-specific stimulus decoding accuracy in populations of neurons along the stimulus axis over time prior to the go cue. Shaded areas indicate mean ± SEM. Bottom. Quantification of region-specific stimulus decoding accuracy in populations of neurons projected to the stimulus axis (***P < 0.001, **P < 0.01, *P < 0.05, ALM: 8, 9, 9; M1a: 10, 12, 12; M1p: 11, 13, 13; S1fl: 10, 12, 11; vS1: 10,

12, 12; M2: 12, 14, 14; RSC: 12, 14, 14; PPC: 12, 14, 14 sessions from 7 mice for naive, intermediate and expert, respectively, one-tailed bootstrap with FDR). Error bars indicate mean ± SEM. **b** Same as (**a**) for choice (***P < 0.001, ALM: 7, 9, 9; M1a: 8, 12, 12; M1p: 9, 13, 13; S1fl: 8, 12, 11; vS1: 8, 12, 12; M2: 10, 14, 14; RSC: 10, 14, 14; PPC: 10, 14, 14 sessions from 7 mice for naive, intermediate and expert, respectively, one-tailed bootstrap with FDR). Error bars indicate mean ± SEM.

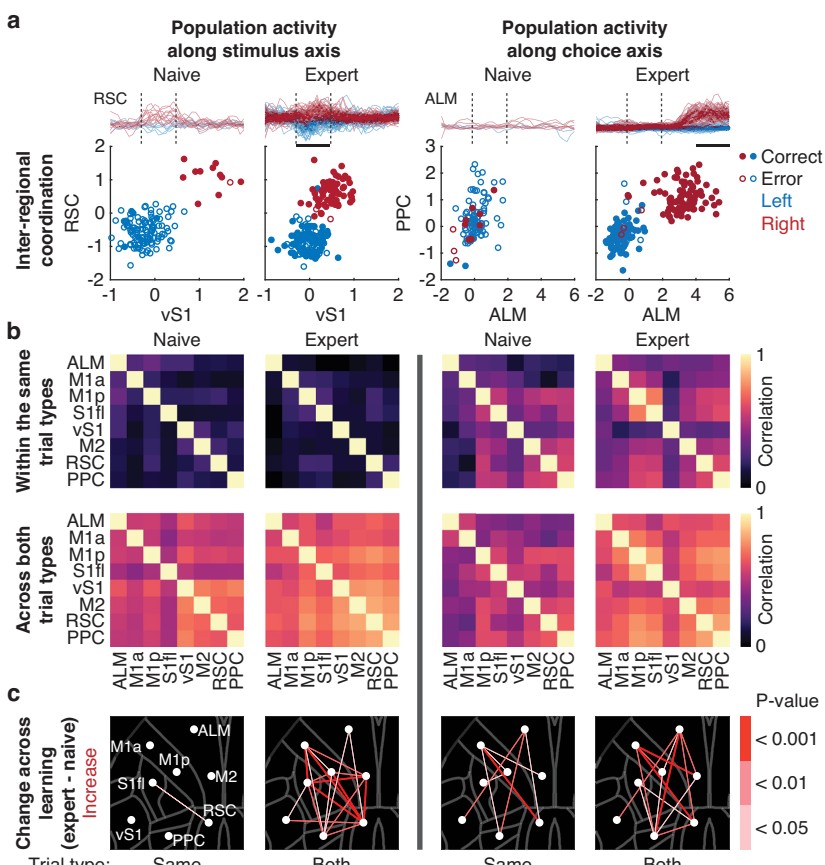

**Fig. 3 | Coordination of sensorimotor representation across learning. a** Example sessions showing the emergence of correlated population activity between pairs of regions across learning. Each point denotes averaged activity during the stimulus or pre-action epoch (black horizontal bars) for left and right trials. **b** Top. Trial-by-trial correlations of population activity within the same trial types along the stimulus (left) and choice (right) axis across learning. Bottom. Same as above but for both trial types. **c** P-values for increases in the correlation coefficients across learning in (**b**) with respect to the location of each cortical region (one-tailed bootstrap). Same: same trial types; Both: both trial types.

full model: naive: $0.14 \pm 0.01$; intermediate: $0.15 \pm 0.01$; expert: $0.14 \pm 0.02$, mean $\pm$ SEM, $P = 0.16$, $n = 13$ sessions from 7 mice, repeated measures one-way ANOVA, Fig. S4b). Using GLM, each neuron was further classified as stimulus-, delay- and action-representing cell, based on its encoding properties (Fig. 4b, c, Fig. S4c and d).

To probe functional connectivity of the dorsal cortex, we extracted statistically significant functional coupling between neurons. The significant functional coupling generally reduced over learning, as illustrated by the changes in connectivity matrices (Fig. 4d and Fig. S5a). To quantify this, we computed the convergence index, which describes the fraction of neurons in a source region that were functionally connected to each neuron in a target region (Fig. 4e). Across learning, the convergence index gradually reduced ($P < 0.001$, $n = 1475$ neurons from 7 mice, one-way ANOVA, Fig. 4e). Region-by-region analysis revealed learning-dependent sparsening of functional coupling within and across cortical regions (Fig. 4f and g). Notably, the intra-regional convergence index was persistently higher than the inter-regional convergence index throughout learning, which is consistent with the mesoscale synaptic connectivity of the mouse cortex[61] ($P < 0.001$ for the main effects of intra- versus inter-region, learning, and cortical regions, three-way ANOVA, Fig. 4f). Sparsening of cortical functional connectivity was further confirmed with pairwise correlations of spontaneous activity of single neurons during ITIs and trial-by-trial pre-action activity (Fig. S5b-f). We confirmed that the effect was not due to trial-type imbalance (Fig. S5g). Thus, the GLM analysis yielded consistent results with more conventional methods to probe functional connectivity (Fig. S5h).

## Selection of behaviorally relevant functional coupling across learning

Given the learning-related augmentation of choice selectivity and global sparsening of functional connectivity in the dorsal cortex, we hypothesized that choice-related functional coupling was selectively retained while choice-irrelevant functional coupling was eliminated. To test this hypothesis, we identified the same population of choice-encoding neurons at the intermediate and expert stages and determined functional coupling that was either retained or eliminated over the two learning stages (Fig. 5a). Consistent with the previous observation, most of the functional coupling was eliminated (retained: $32.2 \pm 2.5\%$; eliminated: $67.7 \pm 2.5\%$, mean $\pm$ SEM, $P < 0.001$, $n = 13$ sessions, one-tailed bootstrap, Fig. 5b).

We reconstructed neural activity at the intermediate stage using retained or eliminated cell-coupling terms in GLM, which were referred to as activity$_{retained}$ or activity$_{eliminated}$. Decoding accuracy for choice was determined at the single neuron level by performing AUC analysis during the pre-action epoch. Decoding accuracy was higher for activity$_{retained}$ than decoding accuracy obtained from activity$_{eliminated}$ in regions such as ALM and PPC (Fig. 5c, $P < 0.05$, one-tailed bootstrap). Session-by-session population activity projected onto the choice axis also confirmed that activity$_{retained}$ had significantly higher choice selectivity than activity$_{eliminated}$ (Fig. 5d, $P < 0.001$, $n = 13$ sessions, one-tailed bootstrap). These results were insensitive to different $\alpha$ values of elastic-net regularization (Fig. S6a and b). Thus, our results suggest that cell coupling in regions such as ALM, which was previously identified to encode choice, was selectively retained while task-irrelevant

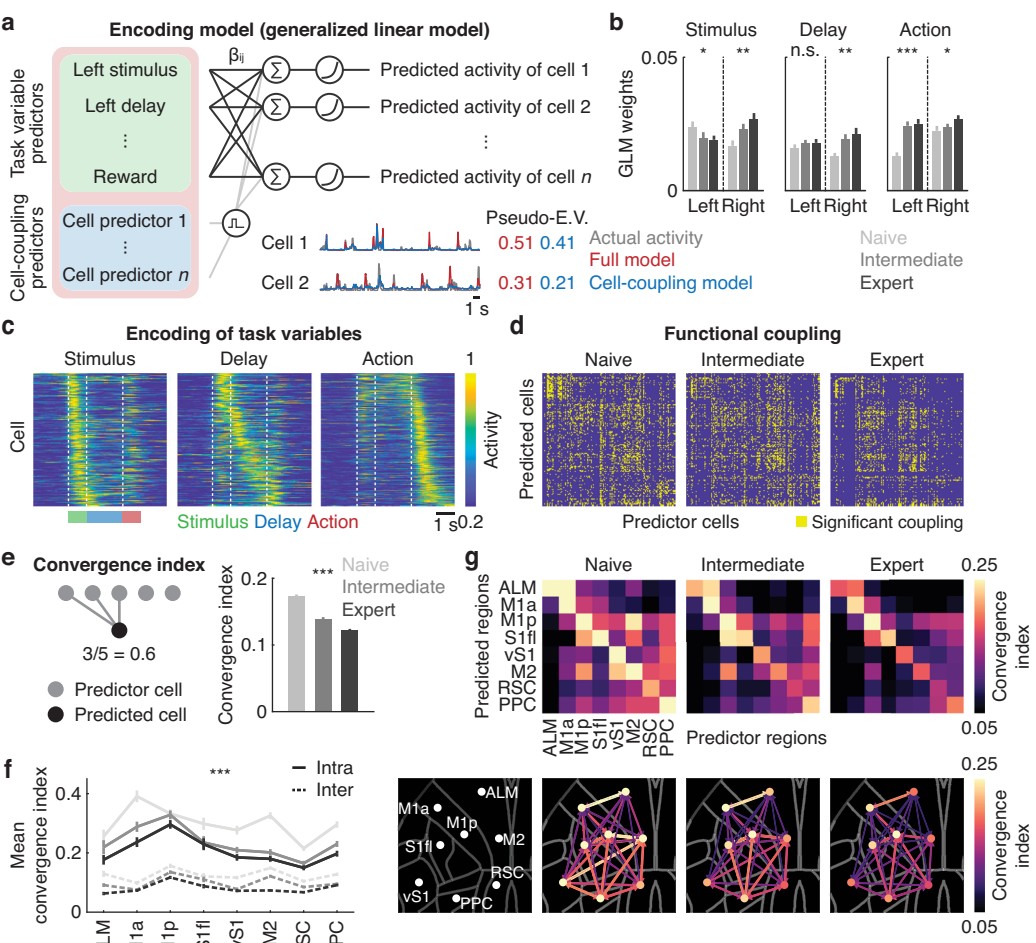

**Fig. 4 | Sparsening of functional connectivity across learning. a** Schematic of generalized linear model (GLM) using task variables and preceding activity of other neurons as predictors. The full model can be decomposed into the cell-coupling model by marginalizing task variable predictors. Activity of two representative neurons reconstructed using the full model (red) and cell-coupling model (blue), in comparison to the actual activity (gray). Numbers indicate the pseudo-explained variance (pseudo-E.V.) of the reconstructed activity. **b** GLM weights for left and right stimulus, delay and action across learning (***$P < 0.001$, **$P < 0.01$, *$P < 0.05$, n.s., $P = 0.17$, $n = 13$ sessions from 7 mice repeated measures one-way ANOVA with FDR). Error bars indicate mean ± SEM. **c** Activity of right-preferring neurons in right trials at the expert stage. Neurons were sorted based on the activity timing in one half of the trials and displayed using the other half (stimulus: 371; delay: 285; action: 504 neurons from 7 mice. **d** Representative sessions from one mouse showing

global sparsening of functional coupling. Each yellow box denotes significant functional coupling (140 neurons). **e** Left. Schematic of convergence index by quantifying functional coupling between predictor neurons in a source region and a predicted neuron in a target region. Right. Decrease in convergence index in the dorsal cortex across learning (***$P < 0.001$, $n = 1475$ neurons from 7 mice, one-way ANOVA). Error bars indicate mean ± SEM. **f** Changes in intra- and inter-regional convergence index across learning (***$P < 0.001$ for main effects of intra- versus inter-region, across learning and across regions, ALM: 82; M1a: 123; M1p: 211; S1fl: 177; vS1: 205; M2: 94; RSC: 287; PPC: 242 neurons from 7 mice, three-way ANOVA). Color scheme is the same as (**e**). Error bars indicate mean ± SEM. **g** Top. Mean convergence index across the dorsal cortex at the three learning stages. Bottom. Convergence index with respect to the location of each cortical region.

coupling was lost during learning. Because we restricted our analysis to cells that were commonly identified across the three learning stages, the limited number of cell-coupling terms prevented us from determining activity$_{retained}$ and activity$_{eliminated}$ between specific pairs of regions.

## Emergence of selective communication channels across learning

In a delayed-response task, it is considered that persistent activity during short-term memory is generated by recurrent connectivity between neurons with similar tuning properties[36,38,40]. We hypothesized that while the global functional connectivity was reduced during learning, connectivity within a subnetwork supporting the persistent activity was strengthened. To determine the connection probability of similarly tuned neurons, we introduced an enrichment index, a fraction of functional couplings between similarly tuned neurons among all functional couplings (Fig. 6a). Across learning, the enrichment

index for choice increased significantly, indicating that choice-encoding neurons became more coupled with other choice-encoding neurons ($P < 0.01$, $n = 13$ sessions from 7 mice, one-tailed bootstrap, Fig. 6b). The increased enrichment index for choice could not simply be explained by an increase in movement stereotypy during the pre-action epoch (Fig. S7a) or trial-type imbalance (Fig. S7b). Stimulus-encoding neurons showed a similar trend, but the increase did not reach statistical significance (n.s., $P = 0.08$, $n = 13$ sessions from 7 mice, Fig. 6b).

To investigate whether the increased connectivity between neurons with the same encoding property was related to stimulus and choice selectivity, we performed AUC analysis on the reconstructed activity using the cell-coupling model for each neuron. Potential covariates of learning-related changes in movement were removed by marginalizing activity derived from the movement predictors in the model. Importantly, across learning, choice-encoding neurons with higher enrichment index became predictive of choice selectivity

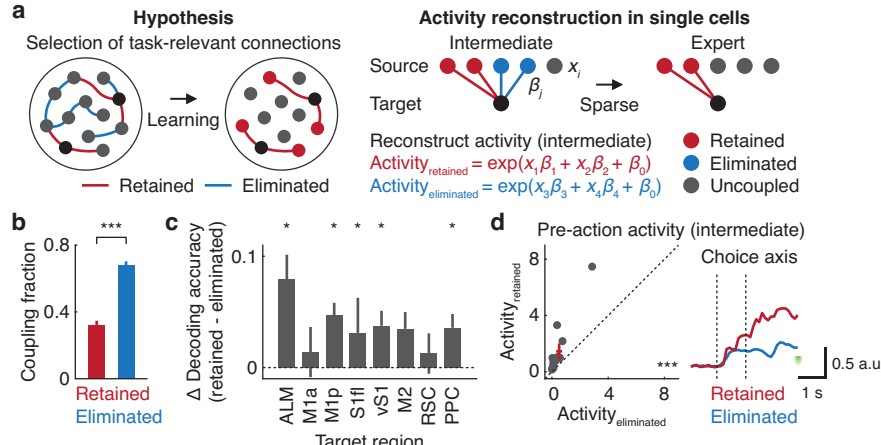

**Fig. 5 | Selection of functional coupling containing choice information across learning. a** Left. Hypothesis of selective retention and elimination of functional coupling across learning. Right. Schematic of functional couplings to the choice-encoding neuron (black) that are either retained (red) or eliminated (blue) across learning. Activity$_{retained}$ and activity$_{eliminated}$ are reconstructed activities with retained and eliminated coupling terms, respectively. **b** Fractions of retained and eliminated functional coupling from the intermediate to expert stage (***$P < 0.001$, $n = 13$ sessions from 7 mice, one-tailed bootstrap). Error bars indicate mean ± SEM. **c** Differences in decoding accuracy for individual choice-encoding neurons

between activity$_{retained}$ and activity$_{eliminated}$ (*$P < 0.05$, ALM: 26; M1a: 42; M1p: 129; S1fl: 104; vS1: 85; M2: 24; RSC: 76; PPC: 105 neurons from 7 mice, one-tailed bootstrap). Error bars indicate mean ± SEM. **d** Choice selectivity computed by averaging pre-action population activity projected to the choice axis (1 s in duration, starting 1 s prior to the go cue, ***$P < 0.001$, $n = 13$ sessions from 7 mice, one-tailed bootstrap). Red cross indicates mean ± SEM. Inset: session-averaged reconstructed population activity prior to the go cue with retained (red) or eliminated (blue) coupling projected to the choice axis.

(Fig. 6c, naive: $R^2 = 0.0016$, n.s., $P = 0.14$, $n = 1017$ neurons from 7 mice; expert: $R^2 = 0.10$, $P < 0.001$, $n = 1154$ neurons from 7 mice, Pearson correlation). Conversely, such learning dependency was weak for stimulus-encoding neurons; the positive relationship was already present at the naive stage in many cortical regions except ALM, M2 and RSC, with the slope becoming only slightly steeper at the expert stage (Fig. 6c and Fig. S7c, naive: $R^2 = 0.026$, $P < 0.001$, $n = 1050$ neurons from 7 mice; expert: $R^2 = 0.078$, $P < 0.001$, $n = 1221$ neurons from 7 mice, Pearson correlation). Together, these results reveal that within a sparsely connected global network of the dorsal cortex, there emerged choice-related subnetworks composed of strongly coupled neurons.

We next examined changes in intra- and inter-regional enrichment indices for stimulus- and choice-encoding neurons. While most regions experienced a decrease in functional coupling over learning, enrichment indices of choice-encoding neurons increased across learning between regions such as ALM and PPC (Fig. 6d, $P < 0.05$, one-tailed bootstrap). In particular, choice-encoding neurons increased mutual connectivity between ALM and PPC and formed a recurrent subnetwork (Fig. 6d). These findings were corroborated with the increased population activity coordination along the choice axis between the two regions (Fig. 3b and c). The observed changes were not simply due to variations in the fractions of stimulus- and choice-encoding neurons, as these changes were more than what would be expected by chance based on randomly sampled functional coupling between neurons (one-tailed bootstrap, Fig. S7d and e). Furthermore, reduced-rank regression analysis for populations of neurons between source and target cortical regions demonstrated a learning-dependent increase in mutual PPC-ALM interactions (Fig. S8a-c). These results suggest that recurrent positive feedback loops between specific regions may be formed to sustain short-term memory.

How does learning modify influences of neural activity in a source area on choice-related activity in a target area? We reasoned that the functional coupling strengthened during learning was critical and their selective "ablation" would reduce choice-related activity in the target area. Since manipulations of such specific functional coupling are experimentally challenging, we selectively removed significant cell-coupling terms in GLM (Fig. 6e). After ablating functional coupling from a source region of interest, we reconstructed neural activity in the target

region using remaining functional coupling and projected the resulting population activity onto the choice axis. Control neural activity was also reconstructed by ablating the same number of functional coupling terms from other source regions. Ablation of PPC-ALM functional coupling, in particular, led to a significant reduction of choice-related activity in ALM (Fig. 6f, g and Fig. S9a, $P < 0.05$, one-tailed bootstrap with FDR). The difference across regions was not due to the different numbers or fractions of cell-coupling terms considered to reconstruct the choice activity (Fig. S9b). Furthermore, the replacement of pre-action activity with scrambled ITI activity yielded similar results (Fig. S9c and d). These results demonstrate that the learning-related emergence of selective communication channels is critical for forthcoming choice.

## Task performance and ALM activity shaped by PPC

To further elucidate the PPC-ALM communication in the delayed-response task, we next examined how PPC activity was related to the task performance and ALM activity. To this end, we performed two complementary experiments. First, we optogenetically silenced PPC using transgenic mice expressing Channelrhodopsin-2 (ChR2) in GABAergic neurons and evaluated its behavioral consequence (Fig. 7a). This approach enabled temporally restricted suppression of PPC activity during the delay period. Photoinhibition of PPC, but not vS1, led to deterioration in the task performance in expert mice (PPC: $P = 0.005$, $n = 15$ sessions from 5 mice; vS1: $P = 0.36$, $n = 15$ sessions from 5 mice, one-tailed bootstrap, Fig. 7b), indicating that PPC was crucial for the task. This observation, however, seemingly contradicted with a previous study with a similar behavioral paradigm[35]. We hypothesized that the discrepancy was due to the difference in training duration and predicted that the contribution of PPC would become gradually smaller after extensive training. Consistent with this view, PPC suppression did not affect the task performance when mice were over-trained for an additional 18-50 sessions (PPC: n.s., $P = 0.34$, $n = 12$ sessions from 4 mice; vS1: n.s., $P = 0.23$, $n = 12$ mice, one-tailed bootstrap, Fig. 7b). These results suggest that PPC was involved in a relatively early phase of learning.

Second, we used designer receptors exclusively activated by designer drugs (DREADDs)[62] by injecting AAV8-CaMKIIα-hM4D(Gi)-mCherry into PPC and suppressed its activity while recording the

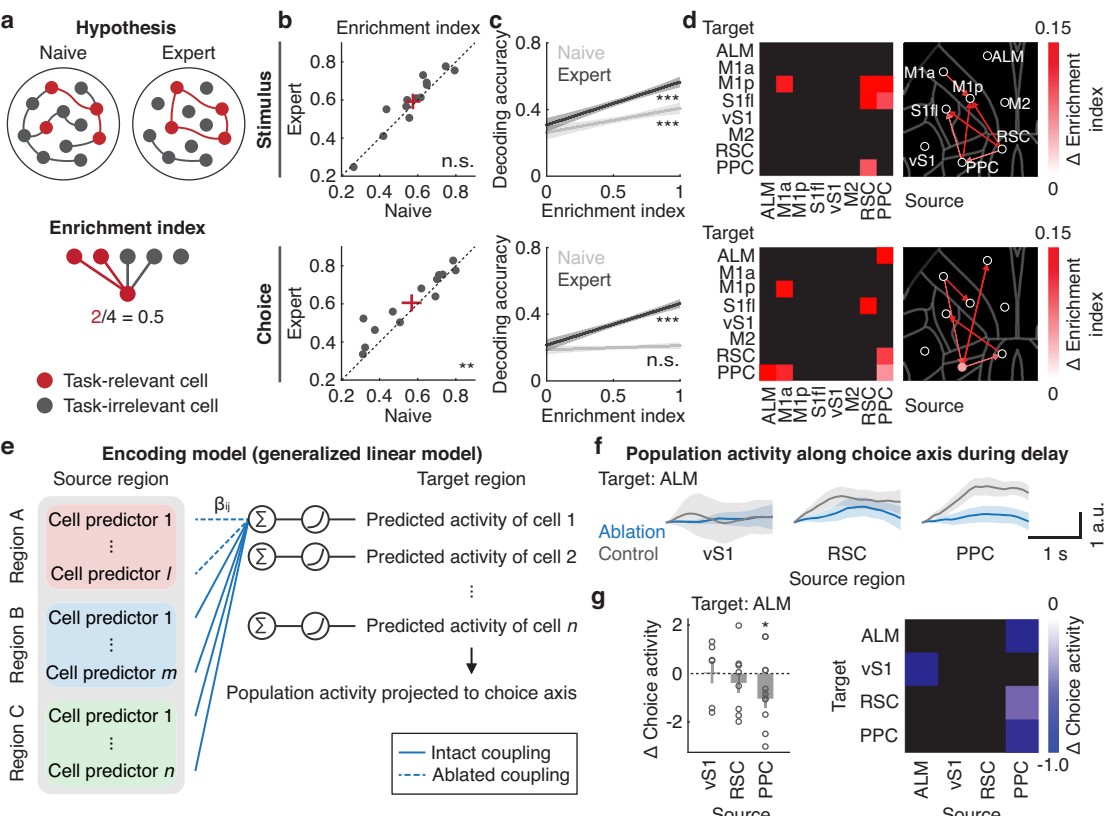

**Fig. 6 | Emergence of PPC-ALM subnetwork across learning. a** Top. Hypothesis of changes in cortical network motifs across learning. Bottom. Schematic of enrichment index. **b** Changes in enrichment index of stimulus- (top) and choice- (bottom) encoding cells across learning (stimulus: n.s., $P = 0.08$; choice: **$P < 0.01$, $n = 13$ sessions from 7 mice, one-tailed bootstrap). Red crosses indicate mean ± SEM. **c** Relationship between enrichment index and decoding accuracy of stimulus (top) and choice (bottom) in single neurons determined by AUC analysis (stimulus: ***$P < 0.001$, $n = 1050$ and 1221 neurons from 7 mice; choice: n.s., $P = 0.14$ and ***$P < 0.001$, $n = 1017$ and 1154 neurons from 7 mice for naive and expert, respectively, Pearson correlation). Shaded areas represent 95% confidence intervals from the mean. **d** Left. Learning-related increases in enrichment index for right-preferring stimulus (top)- and choice (bottom)-encoding cells after accounting for random changes in functional coupling (**Fig. S7d** and **e**). Only pairs of regions with $P < 0.05$ (one-tailed bootstrap) are displayed. Right. Changes in enrichment index with respect to the location of each region. Color of solid circles indicates intra-regional enrichment index. Thickness of lines is proportional to inter-regional enrichment index. **e** Schematic of activity reconstruction of target neurons following selective ablation of cell-coupling terms between source and target regions. **f** Session-averaged reconstructed population activity with functional coupling following targeted (blue) and control ablation (gray) of cell-coupling terms projected to the choice axis during the delay epoch (vS1: 6; RSC: 8; PPC: 9 sessions from 6 mice). Shaded area indicates mean ± SEM. **g** Left. Differences between choice-related population activity following targeted and control ablations averaged within the pre-action epoch (*$P < 0.05$, vS1: 6; RSC: 8; PPC: 9 sessions from 6 mice, one-tailed bootstrap with FDR). Error bars indicate mean ± SEM. Right, Same as left but for additional pairs of regions that showed statistically significant changes (one-tailed bootstrap).

activity of ALM neurons with two-photon calcium imaging (Fig. 7c). While this method lacks the temporal precision of optogenetics, it enabled us to simultaneously image neural activity. We first confirmed that systemic injection of clozapine N-oxide (CNO) into mice expressing hM4D(Gi)-mCherry reduced the activity of PPC neurons (Fig. S10a and b). As in the case of PPC photoinhibition, PPC inactivation reduced the task performance ($P = 0.009$, 18 sessions from 6 mice, one-tailed bootstrap, Fig. 7d). In addition, this manipulation reduced choice selectivity in ALM neurons, which was determined by subtracting the pre-action activity of left choice trials from right choice trials ($P = 0.008$, 16 sessions from 6 mice, one-tailed bootstrap, Fig. 7e). This metric of choice selectivity was correlated with the task performance ($R^2 = 0.22$, $P < 0.01$, $n = 32$ sessions from 6 mice, saline and CNO combined, Pearson correlation). Thus, the PPC-ALM communication was recruited during learning of the delayed-response task and PPC shaped ALM activity for appropriate decision-making.

**PPC-ALM communication channel for stable attractor dynamics**
Improvement of the task performance may result from learning-related enhancement in the robustness of attractor dynamics along the choice code. During a two-alternative forced choice task, for example, population activity in ALM reaches two-point attractors corresponding to two choices, whose basins are separated by a saddle point[38,39] (Fig. 8a). We hypothesized that the separation and depth of the basins were subject to extrinsic inputs, specifically those derived from PPC. To explore conditions that render choice-related cortical activity more robust, we built recurrent neural networks (RNNs) using a target-based recursive least-square algorithm (FORCE)[63,64] to mimic the activity of individual ALM neurons ($n = 807$ neurons from 6 mice, Fig. 8b, c, Fig. S11a and b). These neurons were recurrently connected to external units mimicking the activity of an external region to ALM ($n = 128$ units, Fig. 8b). Based on the ablation study with GLM (Fig. 6e-g and Fig. S9a), we selected PPC and vS1 as external regions of interest and hypothesized that they would distinctively influence the ALM choice selectivity. By projecting the population activity of ALM units onto the choice axis, we determined the frequency of decision switching in response to perturbation, presented for 500 ms after the offset of the stimulus (Fig. 8b and d). Consistent with our hypothesis, the addition of reconstructed PPC-ALM activity significantly improved the stability of the attractor states compared to the addition of vS1-ALM activity ($P < 0.001$, $n = 21$ RNNs, one-tailed bootstrap, Fig. 8d, e and Fig. S11c).

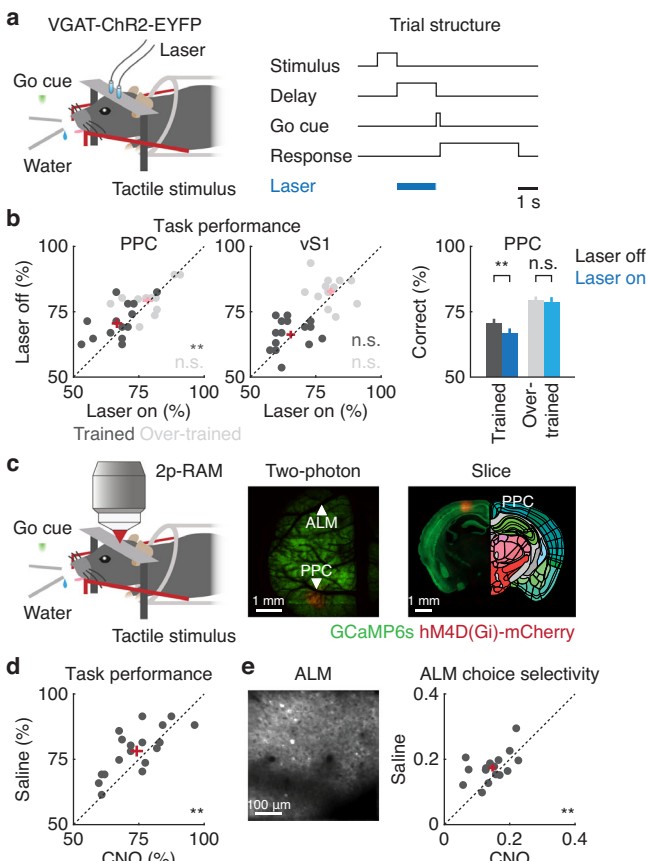

**Fig. 7 | Task performance and ALM activity shaped by PPC. a** Schematic of the optogenetics experiment with VGAT-ChR2-EYFP transgenic mice. Laser was turned on during the delay period in randomly interleaved trials. **b** Left. Task performance when PPC or vS1 was photoinhibited in normally trained or over-trained mice (PPC: normally trained: **P = 0.005, n = 15 sessions from 5 mice; over-trained: n.s., P = 0.34, n = 12 sessions from 4 mice; vS1: normally trained: P = 0.36, n = 15 sessions from 5 mice; over-trained: n.s., P = 0.23, n = 12 sessions from 4 mice, one-tailed bootstrap). Red crosses indicate mean ± SEM. Right: Summary data of task performance with PPC photoinhibition (Same as left). Error bars indicate mean ± SEM. **c** Schematic of the designer receptors exclusively activated by designer drugs (DREADDs) experiment and injection site of AAV8-CaMKIIα-hM4D(Gi)-mCherry. **d** Task performance when PPC was silenced with clozapine N-oxide (**P = 0.009, 18 sessions from 6 mice, one-tailed bootstrap). Red cross indicates mean ± SEM. **e** Left. Example image of ALM neurons expressing GCaMP6s obtained from (**c**). Right. Mean ALM choice selectivity in choice-encoding neurons, derived by subtracting left choice activity from right choice activity during the pre-action epoch (**P = 0.008, 16 sessions from 6 mice, one-tailed bootstrap). Red cross indicates mean ± SEM.

These results remained generally constant across different hyper-parameter configurations (Fig. S11c and d).

To determine if the robustness of attractor dynamics was influenced by the recurrency between PPC and ALM, we randomly ablated their connections (Fig. 8f). Frequency of decision switching was significantly augmented as the fraction of ablated connections increased (*P* = 0.005, *n* = 21 RNNs, one-way ANOVA, Fig. 8g and Fig. S11c). Furthermore, the robustness of attractor states in PPC-ALM was dependent on learning (*P* < 0.001, naive: *n* = 20; expert: *n* = 21 RNNs one-tailed bootstrap, Fig. 8h). The increased robustness could partially be due to the increase in choice-selective inputs inherited from PPC (Fig. S11e and f). Taken together, these results suggest that learning enhances the robustness of cortical dynamics through the recruitment of the long-range and recurrently connected PPC-ALM communication channel, rendering the choice-related attractor dynamics more stable (Fig. 8i).

## Discussion

While learning is considered to involve brain-wide activity modulations of individual neurons and reorganization of functional neural networks, there is little evidence supporting this view. Using a 2p-RAM, we longitudinally imaged the activity of the same population of neurons spanning across eight cortical regions during learning of a delayed-response task. Such mesoscale imaging with cellular resolution enabled us to discover the emergence of subnetworks across learning, where functional coupling between choice-encoding neurons was strengthened in a globally sparsening functional network. Such reorganization of the functional circuit in the dorsal cortex was under a selective pressure where behaviorally relevant functional coupling was selectively retained across learning. As similar learning-related refinement of functional connectivity within a single region has been reported to occur during sleep[65], we predict that sleep contributes to the emergence of the observed cortical network motifs.

ALM and PPC have been investigated in isolation with similar two-alternative forced-choice tasks in rodents[35,38,41,48]. While we confirmed that similar coding properties existed in each area, a range of analyses pointed to a unified scheme where ALM and PPC became part of a functional subnetwork via reciprocal interactions to support short-term memory. Across learning, neurons in ALM and PPC became more coordinated on a trial-by-trial basis, and functional connectivity between ALM and PPC neurons sharing similar choice representation was more enriched. Simulations with RNNs further confirmed the importance of the PPC-ALM subnetwork for maintaining persistent activity for forthcoming choice and its robustness to perturbation. In contrast, the vS1-ALM subnetwork, whose enrichment index failed to increase, was relatively dispensable for the robust choice code.

During learning, PPC served as an important hub in the functional network. We confirmed the functional importance of PPC reported in previous studies[21,66] and demonstrated that ALM activity was modulated by PPC. However, PPC gradually became dispensable for the delayed-response task, indicating that the identified functional subnetworks were recruited transiently during learning and other subcortical networks would functionally dominate once learning was complete. Similar disengagement of the cortex over extended training has been documented in other domains of learning, including motor learning[67].

Importantly, our analysis of functional connectivity does not provide direct evidence for causal interactions between neurons; it may reflect common or correlated inputs to neurons. For example, because direct connections from PPC to ALM seem to be rare[68,69], PPC-ALM connections are unlikely to be a result of direct cortico-cortical interactions; the dependency of ALM activity on PPC suggests that PPC communicates with ALM through indirect pathways via other subcortical structures[33]. Besides the PPC-ALM mutual connectivity, we also identified PPC-RSC connectivity as another potentially important communication channel for choice representation. While reciprocal connections between PPC and RSC were previously described[68], their contributions to short-term memory had not been extensively studied. On the other hand, while PPC projects heavily to M2[70], their interactions were not strong during the delayed-response task. Thus, our approach highlighted the importance of evaluating functional subnetworks that could not be solely explained by anatomical connectivity. Future investigation may involve precise mapping and manipulation of the selected connectivity to determine how such subnetworks form during learning.

While the focus of the present study was choice representation during short-term memory, we also observed some changes in stimulus representation across learning. Generally, while learning-related changes in stimulus representation were relatively small, decoding accuracy for the stimulus location was enhanced in regions such as M2,

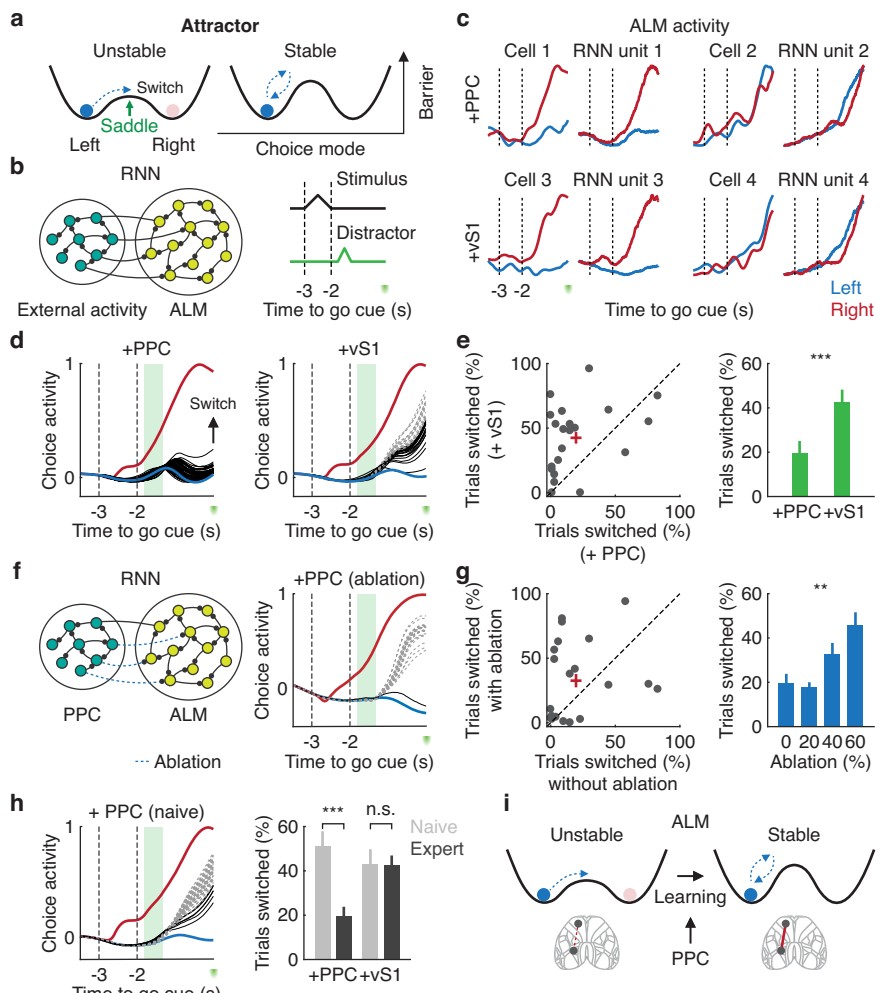

**Fig. 8 | Enhanced robustness of the choice code in ALM by PPC. a** Schematic of attractor dynamics along the choice axis during the delay. **b** Schematic of the RNN architecture trained to mimic the activity of ALM neurons. A stimulus pulse was delivered to the RNN instructing the right trial type. In perturbation trials, a distractor was delivered to the trained RNN. **c** Example RNN units (right) reproducing the activity of ALM neurons (left) trained either with PPC-ALM (+PPC) or vS1-ALM (+vS1) external activity. **d** Example population activity of ALM units projected to the choice axis. Red and blue trajectories indicate average choice activity for right and left trial types without a distractor, respectively. Dashed gray and solid black trajectories indicate individual perturbation trials that did and did not switch decision, respectively. Shaded green rectangle denotes the distraction epoch. **e** Left. Pairwise comparison of fractions of decision-switching trials between RNNs trained with PPC and vS1 ($n = 21$ RNNs). Red cross indicates mean ± SEM. Right. Fractions of decision-switching trials between RNNs trained with PPC and vS1 ($n = 21$ RNNs, ***$P < 0.001$,

one-tailed bootstrap). Error bars indicate mean ± SEM. **f** Left. Schematic of an RNN similar to (**b**) but with random ablation of connections between ALM and external units from PPC. Right. Population activity of an example RNN trained with PPC-ALM external units after 40% ablation. **g** Left. Pairwise comparison of fractions of decision-switching trials between RNNs with and without 40% ablation ($n = 21$ RNNs). Red cross indicates mean ± SEM. Right. Fractions of decision-switching trials as a function of percentage of connections ablated ($n = 21$ RNNs, **$P = 0.005$, one-way ANOVA). Error bars indicate mean ± SEM. **h** Left. Same as (**d**) but with an example RNN trained with the same ALM units but using reconstructed PPC-ALM external activity from a naive session. Right. Fractions of decision-switching trials between RNNs across learning (+PPC: ***$P < 0.001$; +vS1: n.s., $P = 0.46$, naive: $n = 20$; expert: $n = 21$ RNNs, one-tailed bootstrap). Error bars indicate mean ± SEM. **i** Schematic illustrating how PPC influences the choice-related attractor dynamics in ALM.

consistent with previous studies[52,53]. Thus, while major changes occurred in choice representation, stimulus information also emerged in some cortical regions across learning.

A natural extension of the current study is to investigate brain-wide reorganization of the functional network during learning. A new technology such as Neuropixels 2.0 allows longitudinal and simultaneous recordings from the same population of neurons in cortical and subcortical regions[14], which will complement recent studies on cortical-subcortical functional connectivity[19]. We predict that the principles discovered in our study–global sparsening of functional connectivity and the emergence of subnetworks with similar functional relevance–are generalizable to brain-wide functional networks. Our analytical approach may reveal previously unseen functional connectivity beyond short-term memory.

## Methods

### Animals

All procedures were in accordance with the Institutional Animal Care and Use Committee at Nanyang Technological University. Transgenic mice were obtained from the Jackson Laboratory (CaMKII-tTA: 007004; TRE-GCaMP6s: 024742; VGAT-ChR2-EYFP: 014548). Mice were housed in a reversed light cycle (12 h:12 h) in standard cages at around 21 °C and 62% humidity and experiments were generally performed during the dark period. Both male and female hemizygous mice were used. Sample size was determined based on previous studies[17,39,53].

### Surgery

Adult mice (between 8-week and 4-month-old) were anesthetized with 1-2% isoflurane and a piece of scalp was removed. After the underlying

bone was cleaned with a razor blade, a craniotomy (~7 mm in diameter) was made in the left hemisphere (center of the craniotomy: 0.20 mm anterior and 2.00 mm lateral to the bregma) with a dental drill and an imaging window was placed in the craniotomy. The imaging window was constructed from a small (~6 mm in diameter) glass plug (#2 thickness, Fisher Scientific, 12-540-B) attached to a larger (~8 mm in diameter) glass base (#1 thickness, Fisher Scientific, 12-545-D) using an ultraviolet-curing adhesive (Norland, NOA 61). 1.5% agarose (Sigma-Aldrich, A9793-50G) was applied to fill the gap between the skull and window. A custom-built titanium head-plate was then implanted on the window with cyanoacrylate glue and cemented with black dental acrylic (Lang Dental, 1520BLK or 1530BLK). Buprenorphine (0.05-0.1 mg/kg of body weight), Baytril (10 mg/kg of body weight) and dexamethasone (2 mg/kg of body weight) were subcutaneously injected, and mice were monitored until they recovered from anesthesia.

For surgery of the optogenetics experiment, optic fiber sleeves (Thorlabs, ADAL1) were placed bilaterally over the skull of VGAT-ChR2-EYFP mice with cyanoacrylate glue at PPC (−1.75 mm anterior-posterior, AP, 2.05 mm medial lateral, ML) and vS1 (−1.15 mm AP and 3.45 mm ML). Black dental acrylic was applied after a custom-built aluminum head bar was attached. Buprenorphine (0.05-0.1 mg/kg of body weight) and Baytril (10 mg/kg of body weight) were subcutaneously injected, and mice were monitored until they recovered from anesthesia.

For surgery of the designer receptors exclusively activated by designer drugs (DREADDs) experiment, adeno-associated virus (pAAV-CaMKIIα-hM4D(Gi)-mCherry, addgene, 50477-AAV8) was injected bilaterally in PPC: −1.75 mm AP, 2.05 mm ML at the depth of 200 μm and 500 μm (~50 nl for each site) before the imaging window was implanted. After each injection, the pipette was left in the brain for 4 minutes before it was slowly withdrawn.

## Behavior

Water-deprived mice were trained on a delayed response task (one session per day). Each trial started with a left or right tactile stimulus (~20 Hz sweeping, 1 s in duration) followed by a delay (2 s in duration) and a response (4 s in duration) epoch. The beginning of the response epoch was signaled by a go cue (green LED, 525 nm, Thorlabs LED525E, 0.2 s in duration) and both left and right water spouts moved closer to mice and were made available. Mice were trained to lick one of the water spouts on the same side as the whisker stimulation to receive 4 μl of sucrose water (10-15%) as a reward per trial. No response or incorrect response trials were punished with a white noise and a 6-8-s timeout. If mice lick the same side regardless of the trial type for more than five consecutive trials, the other trial type was selected for the subsequent trial to prevent bias. Each session consisted of 180 trials. Mouse behavior was video recorded at 60 Hz at a resolution of 1040 × 900 pixels using a monochromatic camera (FLIR, BFS-U3-16S2M-CS-SET).

For each mouse, naive, intermediate, and expert sessions (two sessions for each learning stage) were determined. The naive sessions were defined as the first and second sessions by default. However, the first session for one mouse was excluded because there were fewer than five responses (correct or incorrect) for the left trial type. The first sessions for two mice were also excluded due to technical issues encountered during the recording of their behavior while they were properly trained. The expert sessions were defined as the first two sessions where the correct rate consistently reached ~70-75%. Two intermediate sessions were selected from midpoints of the naive and expert sessions. In total, we analyzed 14 sessions from 7 mice at each learning stage. Two mice were excluded because they were unable to reach more than 70% correct rate within 30 sessions.

## Optogenetics

Transgenic mice expressing Channelrhodopsin-2 (ChR2) in GABAergic neurons (VGAT-ChR2-EYFP) were trained with the same protocol

as above but with additional reward trials (20% of total trials), where mice were rewarded at the lick port corresponding to the side of whisker stimulation regardless of their choice. The reward trials were removed after mice reached the criterion of >70% correct rate for two consecutive sessions. One mouse was excluded as it was unable to reach the criterion within 30 sessions. Once the criterion was reached, an additional trial type was introduced where a tactile stimulus was presented to both sides of the whiskers for 0.2 s during the delay epoch (starting at 0.25 s after the stimulus offset, ~50% of total trials) as a distractor. Approximately 50% of randomly chosen trials were coupled with optogenetic inhibition, where a 1.9 s laser square pulse followed by a 0.1 s taper (473 nm, power of ~2.5 mW/mm², Shanghai Laser and Optics Century, BL473T8-100FC) was delivered bilaterally through the sleeves attached to the skull during the delay period. Each mouse performed three rounds of PPC and vS1 photo-inhibition sessions (two sessions per round, six sessions in total), where the order of these sessions in each round was randomized (PPC: $n = 15$; vS1: $n = 15$ sessions from 5 mice). Only trials without a distractor were further analyzed.

After six optogenetic sessions, mice continued to be trained with the same protocol for additional 18–50 sessions. Mice that reached 75% correct rate for two consecutive sessions were deemed to be overtrained and tested for another six sessions of photoinhibition (PPC: $n = 12$; vS1: $n = 12$ sessions from 4 mice). One mouse was excluded as it was unable to reach this criterion within 50 sessions.

## Two-photon calcium imaging

Calcium imaging data were acquired using a two-photon random access mesoscope (2p-RAM, Thorlabs)[11] controlled with ScanImage (Vidrio Technologies) with a laser (InSight X3, Spectra-Physics) whose excitation wavelength was tuned to 940 nm with the power of ~40 mW at the objective lens. The imaging frame rate was ~9.35 Hz and the imaging resolution was 1 × 0.4 pixel/μm with eight fields of view (FOVs) of 500 × 500 μm, which were imaged simultaneously at the depth of ~150–200 μm. The stereotaxic coordinates for these eight FOVs relative to the bregma were: ALM: 2.25 mm AP, 1.65 mm ML; M1a: 1.65 mm AP, 2.75 mm ML; M1p: 0.65 mm AP, 1.75 mm ML; S1fl: 0.25 mm AP, 2.65 mm ML; vS1: −1.15 mm AP, 3.45 mm ML; M2: 0.50 mm AP, 0.45 mm ML; RSC: −1.25 mm AP, 0.55 mm ML; PPC: −1.75 mm AP, 2.05 mm ML. The same FOVs were identified in every session and imaged longitudinally.

## DREADDs

Transgenic mice (CaMKII-tTA × TRE-GCaMP6s) were trained with the same protocol as the optogenetics experiment. Once the correct rate reached >70% for two consecutive sessions, the PPC suppression experiment commenced. One mouse was excluded as it was unable to reach the criterion within 30 sessions. Each mouse performed three rounds of clozapine N-oxide (CNO) and saline sessions (two sessions per round, six sessions in total), where the order of these sessions in each round was randomized. We excluded four imaging sessions (two CNO sessions and two saline sessions) for one mouse due to occlusion of the optical window (CNO: $n = 16$; saline: $n = 16$ sessions from 6 animals). CNO (Sigma-Aldrich, C0832-5MG) was dissolved in dimethyl sulfoxide (DMSO, Sigma-Aldrich, D2438) to a stock solution of 0.4 g/ml, which was stored at 4 °C. Before each experiment, a working solution of CNO was prepared by diluting the stock solution with saline (0.9% NaCl solution) to a concentration of 0.2 mg/ml. CNO (1 mg/kg of body weight) or saline was intraperitoneally administered to each mouse ~40 minutes before the experiment. Only trials without a distractor were further analyzed.

To study the inhibitory effect of DREADDs, saline was first injected intraperitoneally to each mouse ~40 minutes before the spontaneous activity of PPC neurons was imaged (FOV: 500 × 500 μm; duration: ~8 min; frequency: ~9.35 Hz). Approximately 40 min after imaging with

saline, CNO was injected and spontaneous activity of the same PPC neurons was imaged.

## Analysis

**Data pre-processing.** For body movement data, $x$ and $y$ coordinates of the forelimbs of each mouse were determined with DeepLabCut[71] from the video recorded in each session (60 Hz). To train the neural network, 24 randomly selected 5-minute videos from seven mice were used. From each five-minute video, 20 frames containing a variety of postures were extracted using K-means clustering. Mouse forelimbs were manually labelled using the 20 frames per video for a total of 480 frames, which were then split into 456 training and 24 test frames. The trained network showed a train error of 2.24 pixel and a test error of 12.46 pixel. This trained network was used to estimate the $x$ and $y$ coordinates of the forelimbs across all sessions and mice. We removed data points with less than 95% likelihood and replaced them with an estimate by interpolating the preceding and succeeding frames with more than 95% likelihood. Outlier datapoints defined as larger than three times the median absolute deviation within a 5-frame window were replaced with a previous data point. Time-series of the resulting $x$ and $y$ coordinates was smoothed using a 3-frame sliding window.

For imaging data, each cortical region was pre-processed separately for registration, cell detection and signal extraction using Suite2p[72]. Only cells with deconvolved z-scored activity more than 10 for at least once every 10 minutes were included in the analysis ($n = 14$ sessions, naive: ALM: 1916; M1a: 1822; M1p: 2496; S1fl: 1800; vS1: 1751; M2: 1898; RSC: 2480; PPC: 2375; intermediate: ALM: 1367; M1a: 1388; M1p: 2502; S1fl: 1827; vS1: 1761; M2: 1880; RSC: 2539; PPC: 2263; expert: ALM: 1212; M1a: 1537; M1p: 2629; S1fl: 1602; vS1: 1707; M2: 1592; RSC: 2219; PPC: 2185 neurons). For the same cell analysis the same neurons were registered across naive, intermediate, and expert sessions ($n = 13$ sessions from 7 mice), with an open-source algorithm ROIMatchPub (https://github.com/ransona/ROIMatchPub). The matched cell candidates were manually validated based on cell morphology across sessions (ALM: 86; M1a: 129; M1p: 212; S1fl: 178; vS1: 212; M2: 98; RSC: 312; PPC: 248 neurons). One session with fewer than 25 cells matched across all learning stages was excluded from the analysis.

**Encoding of task variables in neurons.** To identify stimulus-encoding of each neuron, we compared average activity during the stimulus epoch (1 s duration) in the correct left versus correct right trials and determined whether there was a statistical difference (Wilcoxon rank-sum test, $p < 0.05$). To identify choice-encoding, the same procedure was used with average activity during the pre-action epoch (between −1 s and 0 s relative to the go cue).

**Population activity dynamics and dimensionality reduction.** To analyze population activity of $n$ neurons simultaneously recorded in each session, we reduced dimensionality of the $n$-dimensional activity space by projecting the population activity to the stimulus or choice axis in each cortical region. The stimulus axis maximally separated neural trajectories between the left and right trial types during the stimulus epoch, while the choice axis maximally separated neural trajectories between the left and right choice during the pre-action epoch[39]. Regions with fewer than 20 neurons in each session and sessions with fewer than five trials for left and right choice each were excluded from the analysis. To compute the choice mode using the population neural activity, we first computed the vector of average activity differences between the right and left choice trials as follows:

$$\Delta \bar{r} = \bar{r}^R - \bar{r}^L, \tag{1}$$

where $\bar{r}^R$ is trial-averaged activity during the pre-action epoch where mice licked the right water spout regardless of the stimulus. Likewise, $\bar{r}^L$ is trial-averaged activity from trials where mice licked the left water

spout. The resulting $\Delta \bar{r}$ is a weight vector with the size of $n \times 1$. Positive weights were assigned to right-choice-preferring neurons, whereas negative weights were assigned to left-choice-preferring neurons. $\Delta \bar{r}$ was then normalized by its $l^2$ norm to control for the number of neurons that were recorded simultaneously. Projection of the population neural activity along the choice axis, $p^k$, has the size of $t \times 1$ for each trial and was calculated as:

$$p^k = X^k \left( \frac{\Delta \bar{r}}{\sqrt{\sum_i^n |\Delta \bar{r}|^2}} \right), \tag{2}$$

where $k$ and $i$ are the trial and cell index, respectively, and $X^k$ is a matrix of neural activity over time $t$ with the size of $t \times n$.

The stimulus mode was computed similarly but using average activity during the stimulus epoch of left and right stimulus trials regardless of the choice. The choice and stimulus axes were orthogonalized to each other using the Gram-Schmidt process in this order.

For the choice mode, the population activity, $p^k$, was further sorted into left and right choice trials regardless of the stimulus. For the stimulus mode, $p^k$ was sorted into left and right stimulus trials regardless of the choice. Left and right choice or stimulus projected activity was obtained by averaging activity across their respective trial conditions. To compare across sessions, baseline activity was subtracted from trial-averaged projections. The baseline activity was defined as the trial-averaged projection for the respective trial conditions during the ITI epoch (between −1 s and 0 s relative to the stimulus onset).

Choice selectivity of the population activity was determined by computing the difference between right and left activity averaged within the pre-action epoch. The statistical significance of its learning dependency was tested with bootstrap. For each bootstrap, sessions were randomly sampled with replacement for each learning stage and the resulting mean activity during the pre-action epoch was averaged across sessions to compute Pearson correlation coefficients.

**Coordination of population neural activity across learning.** Trial-by-trial coordination of population activity projected to the stimulus or choice axis was computed between pairs of regions. For a given trial, the population activity trajectory was averaged within the stimulus or pre-action epoch. Pearson correlation coefficient was computed with the epoch-averaged activity across trials (either within the same trial conditions or across both trial conditions) between pairs of regions. Statistical significance for the increase in correlations from naive to expert sessions was determined with bootstrap for each region pair. To study a potential confound of the trial-type imbalance, the same number of trials (40 trials) was randomly sampled with the replacement for the left and right trial conditions and the same analysis was performed. This procedure was repeated 100 times and the resulting correlation coefficients were averaged across iterations.

**Generalized linear model (GLM).** Neural encoding of experimentally designed task variables and cell coupling were modelled with the generalized linear model (GLM) for each neuron independently ($n = 1475$ neurons)[44,55,59]. A Poisson GLM was used to compute the weights of predictors in modeling the activity of single neurons based on the deconvolved calcium signal[73]. Variables included the left and right stimulus, left and right delay, left and right lick, left and right forelimb movement, reward and activity of other simultaneously imaged neurons at previous one and two frames. As the task variables were measured at a higher temporal sampling rate (20 kHz) than imaging (9.35 Hz), they were down-sampled by averaging each imaging interval to match the sampling rate of imaging.

The design matrix for GLM was obtained as follows. The stimulus onset times, delay onset times, reward onset times and response onset times were represented as boxcar functions where a value of one was assigned to these times and zero elsewhere. Each task variable was convolved with a set of behaviorally appropriate temporal basis functions to produce task predictors. For stimulus onset times, we used three evenly spaced raised cosine functions extended 1 s forward in time. For delay times, we used five evenly spaced raised cosine functions spanning 1.5 s starting from the end of stimulus offset times. For the reward onset times, we used three evenly spaced raised cosine functions extended 1 s forward in time. For the response onset times, we used five evenly spaced raised cosine functions spanning 1.5 s starting 0.75 s before the end of the delay epoch. For forelimb movement, three evenly spaced raised cosine functions convolved with Euclidean distance moved were used. To examine dependence of the activity of a given predicted neuron on the activity of other simultaneously imaged neurons, additional cell-coupling predictor terms were used. For each predicted neuron, the activity of other neurons was convolved with two boxcar functions ($t+1$ and $t+2$, where $t$ is a given frame) extending -107 ms and -214 ms backward in time from the activity of the predicted neuron.

**GLM fitting.** GLM fitting was performed as described previously[59] after all task and cell-coupling predictors were z-scored. Briefly, the data were first divided into training and test set (70% and 30% of image frames, respectively). The *lassoglm* function in MATLAB with fivefold cross-validation of the training set was used with an elastic-net regularization, which utilizes both lasso and ridge regularization according to the value of a selected parameter $\alpha$, where $\alpha = 0$ corresponds to ridge and $\alpha = 1$ to lasso regression. We used $\alpha = 0.95$ to select a relatively small number of predictors out of many potentially correlated predictors, similar to pure lasso regularization while avoiding potential issues with degeneracies that could arise due to strong correlations between predictors[74]. The number of λ was set at 100. To assess GLM performance, pseudo-explained variance (E.V.) of the model was obtained on the test dataset according to:

$$\text{Pseudo E.V.} = 1 - \frac{D(\hat{y})}{D(\bar{y})}, \tag{3}$$

where

$$D(\hat{y}) = \log L(y) - \log L(\hat{y}) \tag{4}$$

is a deviance from the saturated model in terms of log-likelihoods, whereas

$$D(\bar{y}) = \log L(y) - \log L(\bar{y}) \tag{5}$$

is a deviance from the null model[58]. The null model was calculated with mean activity across frames.

**GLM-derived response profiles.** GLM models neural activity of each neuron by exponentiating the weighted sum of predictors and the estimated neural activity can be decomposed into activity contributed by each predictor[59,75]. The model-derived response profile for a given variable can be defined as a tuning curve for that variable by marginalizing out the effects of other variables. A task variable was defined as statistically significant for the activity of a given neuron when removal of the variable led to a statistically significant decrease in pseudo-E.V by shuffling the task predictors 1000 times using 2-s bins ($P < 0.05$ with FDR). Similarly, a predictor neuron was defined as functionally coupled to a given predicted neuron when removal of either $t+1$ or $t+2$ predictor led to a statistically significant decrease in pseudo-E.V.

The full model for each neuron was reconstructed according to:

$$\hat{y}_{full\,model} = \exp\left(X_{task\,variable}\beta_{task\,variable} + X_{cell\,coupling}\beta_{cell\,coupling} + \beta_0\right), \tag{6}$$

where $\hat{y}$ is reconstructed neural activity for a given neuron, $X_{task\,variable}$ and $X_{cell\,coupling}$ are predictor matrices for task variables and cell coupling, respectively. $\beta_{task\,variable}$ and $\beta_{cell\,coupling}$ are vectors of corresponding coefficients and $\beta_0$ is a bias. For comparisons of GLM weights across learning, the highest value of $\beta_{task\,variable}$ for each task variable was used. For the cell-coupling model, activity was reconstructed using only cell coupling predictors according to:

$$\hat{y}_{cell-coupling\,model} = \exp\left(X_{cell\,coupling}\beta_{cell\,coupling} + \beta_0\right). \tag{7}$$

**Stimulus and choice decoding with receiver operator characteristic analysis.** Stimulus or choice was decoded from single-cell activity, population activity and reconstructed activity using the cell-coupling model. For stimulus, trials were labeled based on the stimulus type regardless of choice. For choice, trials were labeled based on lick direction regardless of the stimulus type. Using either activity from a single frame or average activity during the stimulus or pre-action epoch, we plotted a receiver operator characteristic (ROC) curve and obtained the area under the curve (AUC). The AUC was used to measure the stimulus or choice preference. Decoding accuracy was defined as the absolute deviation of AUC from a chance level according to:

$$\text{Decoding accuracy} = 2^*|\text{AUC} - 0.5|. \tag{8}$$

The relationship between decoding accuracy and enrichment index was examined with Pearson correlation for all neurons across sessions. To identify learning-dependent increases in decoding accuracy, we performed ROC analysis by shuffling trial labels and this procedure was repeated 1000 times to create a null distribution of AUC values. The difference between naive and expert null distribution was obtained and compared with the actual difference to obtain P-values.

**Analysis of convergence and enrichment index with GLM.** Convergence index for each neuron was calculated as:

$$\text{Convergence index} = \frac{n_{cell-coupling}}{n_{total}}, \tag{9}$$

where $n_{Cell-coupling}$ and $n_{Total}$ are the number of statistically significant cell-coupling and all possible cell-coupling, respectively. The convergence indices were averaged across neurons and across sessions for the naive, intermediate and expert stages separately.

Enrichment index for each neuron was calculated as:

$$\text{Enrichment index} = \frac{n_{cell-coupling\,with\,similar\,tuning}}{n_{cell-coupling}}, \tag{10}$$

where $n_{Cell-coupling\,with\,similar\,tuning}$ refers to the number of statistically significant cell-coupling between neurons sharing similar tuning properties (e.g. right-preferring choice-encoding neurons). The enrichment index for a given session and task variable was averaged across neurons.

To calculate the enrichment index between pairs of regions, we analyzed cell-coupling between neurons from a source (predictor) and target (predicted) region. Neurons with consistently zero enrichment index across all learning stages were excluded from the analysis. To assess statistical significance of changes in enrichment index across learning within right-preferring stimulus and choice-encoding neurons, bootstrap was performed. For each bootstrap, neurons were

randomly sampled with replacement for each learning stage and the resulting enrichment indices were averaged. This procedure was repeated 1000 times. An increase in enrichment index from the naive to expert stage was deemed to be statistically significant when the P-value was less than 0.05.

The observed region-specific increases in enrichment index across learning may simply be explained by changes in the number of stimulus or choice-encoding neurons. To confirm that this was not the case, we randomly sampled cell-coupling predictors and computed enrichment index. This procedure was repeated 1000 times. The difference between naive and expert null distributions was obtained for each neuron and compared against the actual change across learning.

**Spontaneous activity and task-related activity correlations.** Correlations of spontaneous activity of pairs of neurons were obtained with Pearson correlation for ITI activity (between 2 s prior to the stimulus onset and the stimulus onset). Pairwise correlations with $P < 0.0001$ were deemed to be functionally connected and the resulting connectivity was used to compute convergence and enrichment index similarly to GLM. Correlations of task-related pre-action activity (between 1 s prior to the go cue onset and the go cue onset) of pairs of neurons were obtained similarly.

To study a potential confound of trial-type imbalance, the same number of trials (40 trials) was randomly sampled with replacement for the left and right trial types and the same analysis was performed to compute the convergence index and enrichment index. This procedure was repeated 100 times and the resulting values were averaged across iterations.

**Analysis of retained and eliminated functional coupling.** For each neuron in a given target region, we identified two types of cell-coupling: (1) retained cell-coupling defined as the cell-coupling that was present at both the intermediate and expert stages and (2) eliminated cell-coupling defined as the cell-coupling that was present at the intermediate stage but not the expert stage. Partial cell-coupling models were reconstructed with either of the two cell-coupling types and referred to as activity$_{retained}$ and activity$_{eliminated}$. To ensure that the same number of cell-coupling terms was used to reconstruct activity$_{retained}$ and activity$_{eliminated}$, the smaller number of retained and eliminated cell-coupling terms was subsampled. Coupling fractions for single neurons were calculated by obtaining the number of retained or eliminated cell-couplings over the total number of statistically significant cell-couplings. The coupling fractions were averaged across neurons.

**Analysis of movement stereotypy and neural activity across learning.** To quantify changes in body movement across learning, trial-averaged Euclidean distance in pixels travelled by the right forelimb during the stimulus or pre-action epoch was computed.

Changes in movement stereotypy were calculated by subtracting the baseline position of the right forelimb, defined as its $x$- and $y$-coordinates at the beginning of each trial, from the coordinates of each frame within the trial. The resulting trajectories during the stimulus or pre-action epoch were used to compute trial-by-trial correlations for the left and right trial types separately.

To determine the relationship between movement stereotypy and intra- and inter-regional coordination of population activity, Pearson correlation was performed between changes in median trial-by-trial correlation coefficients of right forelimb movement and coordination along the stimulus or choice axis within the same or across both trial types in each region pair. To determine the relationship between movement stereotypy and enrichment index, Pearson correlation was performed between changes in median trial-by-trial correlation coefficients of right forelimb movement for right or both trial types and changes in enrichment index in each region pair.

**Reduced rank regression.** To study intra- and inter-regional communications, reduced rank regression was performed. This analysis relies on multivariate linear regression to predict population neural activity of a target region given population neural activity of a source region[76]. As prediction performance of each model was positively related to pairwise correlations between neurons and because the pairwise correlations decreased across learning, the learning-related changes in the prediction performance could be underestimated without a correction. Thus, we matched pairwise correlations of ITI activity of neurons between the naive and expert stages by removing neurons with mean Pearson correlation coefficients, averaged across all pairs for a given neuron, below 0.0274, such that all regions had non-significant differences between the naive and expert stages. Twenty neurons were randomly sampled from source and target regions to compute the prediction performance. This procedure was repeated 20 times and the resulting values were averaged across iterations.

**Analysis of selective ablation of cell-coupling.** To evaluate the effect of ablating a specific group of cell-coupling terms, we reconstructed activity using specific cell-coupling terms between a given pair of regions. We refitted GLM for each neuron at a given expert session without the same cell registration across learning to consider the larger number of predictor neurons. We selected a maximum of 40 neurons from each region for a given expert session and refitted GLM to extract statistically significant functional coupling ($n = 14$ sessions, ALM: 360; M1a: 470; M1p: 520; S1fl: 440; vS1: 480; M2: 555; RSC: 560; PPC: 560 neurons). For each neuron, we reconstructed partial cell-coupling models either with (1) ablation restricted to a specific region (activity$_{ablated}$) or (2) ablation in other regions (activity$_{control}$). For each neuron, ablated coupling was randomly sampled and the partial reconstructed models of a population of neurons were projected to the choice axis. Baseline activity, defined as the activity at the stimulus offset, was subtracted from trial-averaged activity. The resulting activity was normalized by its standard deviation. This procedure was repeated 100 times and the resulting values were averaged across iterations.

To test if choice activity was correlated with the number or fraction of cell-coupling terms used to reconstruct the cell-coupling model, we obtained Pearson correlation coefficient for each session between average choice activity along the choice axis during the pre-action epoch and median number or fraction of cell-coupling terms used to reconstruct activity$_{ablated}$ or activity$_{control}$.

Replacement of the pre-action activity with randomly scrambled ITI activity was performed similarly to the selective ablation described above. For each neuron, activity from a cell-coupling model was reconstructed after replacing the pre-action activity with scrambled ITI activity in a specific region before the same procedure was followed as above.

**Analysis of PPC inactivation with DREADDs.** Imaging data were processed similarly to those obtained for the delayed-response task. Neurons in PPC were registered using the red channel (mCherry) across saline and CNO sessions with Suite2p. For each neuron, deconvolved calcium signal was averaged across imaging frames and compared across the two conditions.

**Recurrent neural network (RNN)**
To study attractor dynamics during short-term memory, recurrent neural networks (RNNs) were built with Python using the first-order reduced and controlled errors (FORCE) algorithm[64]. Units in each RNN were trained to mimic the activity of ALM neurons recorded in the left hemisphere ($n = 807$ neurons) and activity from an external region, PPC or vS1, that was identified in GLM to be functionally coupled to ALM ($n = 128$ neurons). Hyperparameters were optimized by choosing

those that resulted in a good fit for both left and right trial activity. The networks were modeled based on the first-order differential equation as follows:

$$\tau \dot{x}(t) = -x(t) + J \cdot r(t) + W_{stimulus} I_{stimulus}^{k} + W_{cue} I_{cue} + \varepsilon_{noise}(t), \quad (11)$$

where $x$ is the membrane current of the network, $J$ is the recurrent synaptic weight matrix, $r$ is the activity, $W_{stimulus}$ and $W_{cue}$ are the synaptic weight matrices for stimulus and go cue inputs, $I_{stimulus}$ and $I_{cue}$, for the $k$th trial. $J$ was initialized as a square matrix of size $n \times n$, where each element was sampled from the normal distribution, $J \sim \mathcal{N}\left(0, \frac{g}{\sqrt{n}}\right)$. By setting the factor $g > 1$, randomly initialized networks could generate chaotic spontaneous activity prior to training[63]. We initialized $g = 1.1, 1.2, 1.3$ or $1.4$. The initial distribution of $J$ is forgotten after rounds of weight modification. The weight vectors for the stimulus and go cue were sampled from normal distribution $W_{stimulus} \sim \mathcal{N}(0,1)$ and $W_{cue} \sim \mathcal{N}(0,0.1)$. The stimulus input, $I_{stimulus}$, and go cue input, $I_{cue}$, were designed to be in a triangular shape, from $t = -3.0\,s$ to $-2.0\,s$ and $t = -2.1\,s$ to $-2.0\,s$ relative to the action, respectively. The peak amplitude of stimulus for right trials were randomly sampled from a distribution of $I_{stimulus} \sim \mathcal{N}(1.0,0.1)$, while the stimulus input for left trials was fixed at 0. The peak amplitude of $I_{Cue}$ was fixed at 1.0. The noise variable $\varepsilon_{noise} \sim \mathcal{N}(0,0.15)$ was drawn at each time step $t$.

The membrane current $x$ was calculated at each timestep $t$ by integrating the differential equation using Euler method, with the network neural time constant $\tau = 10\,ms$ and integration time constant $\Delta t = 1\,ms$. Activity $r$ was obtained by applying the sigmoidal function to $x$ according to:

$$r(t) = \frac{1}{1 + e^{-\beta(x(t) - \theta)}}, \quad (12)$$

where the parameters were set to be: $\beta = 0.8$ and $\theta = 3.0$. All parameters were identical across all networks, including those trained for activity in naive sessions.

**Training and testing of RNNs.** Each unit in the RNN was trained to reproduce the neural activity of a single ALM neuron, which was computed by averaging activity across correct right choice or left choice trials during expert sessions. The training epoch spanned 3.5 s, starting from −0.5 s from the stimulus onset and ending at the delay offset. For the learning target of the network, ALM neural activity was transformed to the membrane current, $f$, by clipping the neural activity at a maximum value of 5 and minimum value of 0 with slight offsets of −0.01 and 0.01, respectively. The resulting numbers were normalized by a fixed value of 5. At any time step, if unit's normalized activity was lower than $1 \times SD$ across all timesteps and units, it was excluded. This process resulted in a total of 807 neurons.

To compute the external unit activity, PPC-ALM and vS1-ALM, neural activity was reconstructed with partial cell-coupling models using cell-coupling terms from PPC or vS1 to ALM neurons. External activity was similarly normalized to ALM neural activity. For each model, 128 external activities were randomly sampled with replacement. The parameters $\beta = 0.8$ and $\theta = 3.0$ were used to transform the normalized activity data $r$ into target function $f$ with the inverse sigmoidal function:

$$f(t) = \theta + \frac{1}{\beta} \ln\left(\frac{r(t)}{1 - r(t)}\right). \quad (13)$$

To increase the temporal resolution of the network, target function $f$ was up-sampled from sampling rate 9.35 Hz to 93.5 Hz by linear interpolation and was smoothed with a ~400 ms boxcar moving window filter.

Prior to training, a running estimate of the inverse correlation matrix of the network, $P = \alpha \mathbb{1}$, was initialized, where $\mathbb{1}$ is an identity matrix and $\alpha = 0.01$. The learning rate, $\alpha_{learn} = 0.05$, was used for every update. The training phase lasted for 500 epochs for all RNNs. The pseudocode to train the network is:

**Algorithm.** : First Order Reduced and Controlled Errors (FORCE)

Initialize $J, x, W_{stimulus}, W_{cue}, P$
for each training episode do
 Alternate trial type $k \in \{'right', 'left'\}$
 Generate $I_{stimulus}^{k}, \varepsilon_{noise}$
 for each timestep $t$ do
 $z(t) \leftarrow J \cdot r(t) + W_{stimulus} I_{stimulus}^{k} + W_{cue} I_{cue} + \varepsilon_{noise}(t)$
 $x(t) \leftarrow x(t-1) + \frac{\Delta t}{\tau}[-x(t-1) + z(t)]$
 $r(t) \leftarrow \phi(x(t))$
 Calculate error: $e(t) \leftarrow z(t) - f(t)$
 Calculate loss:
$$\Delta J(t) \leftarrow \left[\frac{e(t)}{1 + r^T(t) \cdot P(t-1) \cdot r(t)}\right] \otimes [P(t-1) \cdot r(t)]$$
 Update $J \leftarrow J - \alpha_{learn} \Delta J(t)$
 Update
$$P(t) \leftarrow P(t-1) - \frac{P(t-1) \cdot r(t) \cdot r^T(t) \cdot P(t-1)}{1 + r^T(t) \cdot P(t-1) \cdot r(t)}$$
 end for
end for

Training was designed to allow pairwise comparisons between RNNs trained with PPC-ALM and vS1-ALM external units using the same random seed for weight initialization. Mean squared error (MSE) of each trained RNN was computed by comparing its output to the target activity after averaging it across time of trials and across neurons. To ensure the success of the FORCE algorithm, trained RNNs with MSEs more than $mean_{MSE} + SD_{MSE}$ were analyzed ($n = 21$ RNNs), where $mean_{MSE}$ and $SD_{MSE}$ were computed across all trained RNNs.

The trained RNNs were tasked to generate estimated ALM neural activity in the presence of a distractor lasting 500 ms (mean amplitude = 0.25, 0.30 or 0.35, SD = 0.025) during the early delay epoch (between −1.75 s and −1.25 s relative to the delay offset). Each of the trained RNNs was presented with 100 trials of left, right and left with distractor trial types. For each trial, choice mode was computed similarly to the neural activity of the mouse brain but normalized using the maximum of the right trial choice activity. A left trial with a distractor was considered to have switched the decision if RNN's choice activity at the end of the delay period ended closer to the right trial choice activity in the absence of the distractor, which was defined as above the halfway point between the right and left trial choice activity.

To test if the external activity of PPC-ALM or vS1-ALM was important for the robustness of the choice code, a fraction (20, 40 and 60%) of connections between ALM and 128 external units was randomly ablated, and the frequency of decision switching was computed. This procedure was repeated 100 times for each RNN. To ensure that the change in frequency of decision switching was due to the robustness of the choice code and not due to a poor fit to the target activity, iterations were excluded when the ablations resulted in the choice activity of distractor trials falling below the left choice activity. The results remained similar ($P < 0.05$, one-way ANOVA) without excluding these iterations. Learning dependency of the robustness of the choice code was evaluated using reconstructed external activity obtained from naive sessions ($n = 20$ RNNs).

## Statistics
Statistical tests and error bars are described in relevant sections of the figure legends.

## Reporting summary
Further information on research design is available in the Nature Portfolio Reporting Summary linked to this article.

## Data availability

The processed data are available at Zenodo (https://doi.org/10.5281/zenodo.8031277). Source data are provided with this paper.

## Code availability

The code to generate the main figures is available at GitHub (https://github.com/HiroshiMakinoLaboratory/ChiaEtAl2023Nature Communications). The GitHub repository can be cited with https://doi.org/10.5281/zenodo.8275051.

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

## Acknowledgements

We thank K. Tay and A. Heng for animal husbandry, L. Fontolan for sharing a code for recurrent neural network models and C. Li for comments on the manuscript. This work was supported by NARSAD Young Investigator Grant from the Brain & Behavior Research Foundation (HM), Nanyang Assistant Professorship from Nanyang Technological University (HM), Singapore Ministry of Education Academic Research Fund Tier 1 2018-T1-001-032 (HM) and RT11/19 (HM), Singapore Ministry of Education Academic Research Fund Tier 2 MOE2018-T2-1-021 (HM) and MOE-T2EP30222-0009 (HM) and Singapore Ministry of Education Academic Research Fund Tier 3 MOE2017-T3-1-002 (HM).

## Author contributions

H.M. and X.C. conceived the project. X.C. and L.A. performed the experiments. X.C., H.M. and J.T. analyzed the data. H.M. and X.C. made the figures. H.M. supervised the project. H.M. wrote the manuscript with inputs from X.C. and T.K.

## Competing interests

The authors declare no competing interests.
