## [Peer Review File · Nature Communications]

Emergence of cortical network motifs for short-term memory during learningREVIEWER COMMENTS

Reviewer #1 (Remarks to the Author):

Xin et al. aimed to study communications among neuron ensembles across multiple brain areas during short-term memory. They used 2P-RAM to chronically image populations of neurons in 8 brain areas (ALM, M1a, M1p, M2, S1fl, vS1, RSC and PPC) during the learning process. To analyze the stimulus and choice information during learning, the researchers projected neural activity onto two orthogonal axes that maximized the stimulus and choice selectivity during sample or pre-action epoch, respectively. They found stimulus selectivity remained constant after learning while choice selectivity increased across the dorsal cortex. The correlations between regions along the stimulus axis are relatively low compared the correlations along the choice axis. But the correlations along both axes increase with learning. The main finding of this paper is the cortical functional coupling generally reduced over learning, and this is caused by selective retention of choice-related coupling and elimination of choice-irrelevant coupling. Using a GLM model the authors simulated the choice activity of ALM with or without considering functional coupling of PPC-ALM, RSC-ALM and vS1-ALM. They found that only ablation of PPC-ALM coupling caused a significant reduction of choice related activity in ALM. In addition, after learning, PPC-ALM communication is critical for the robustness of choice-related attractor dynamics.

In summary, the authors used state-of-art meso-2P imaging to track the same population of neurons distributed in multiple regions during learning of a 2AFC task. The authors then used a set of encoding/network simulation methods to analyze the functional coupling during learning and provide some important insights about how long-range functional coupling potentially contributes to neural coding and network robustness. As large scale recording becomes routine and rodents are trained to perform more and more complex tasks, the methods/message conveyed by this manuscript is timely and important. Nevertheless, I have multiple questions/concerns listed below.

Major:

1. It is well known that functional connectivity might reflect common inputs, for example, from more stereotypic movements after learning. I agree that it is experimentally challenging to specifically manipulate functional coupling. Indeed, most optogenetic perturbations have not achieved cellular recapitulation of neural dynamics. However, coarse manipulation often leads to specific effects as behavior/neural dynamics is typically low dimensional. The authors should at least try to perturb PPC and investigate whether this manipulation affects ALM activity as simulated.
2. Related to the above, does PPC project to ALM?
3. Related to 1. The enrichment index and choice activity increased from PPC->ALM. Could this be due to more stereotypic movements. Fig S1 only shows that movements on average did not increase.
4. The authors observed that activity remained encoded choice better than activity eliminated. Could this be due to lasso regularization as this naturally removes components with small contribution?

5. For the encoding model, the explained variance is only about 10%. A naïve question is: what's the meaning of analyzing different components of this 10% variance.

With training, task variables explain larger percentage of explained variance (compared with early training). I am not sure whether this is obvious in Fig. 3b. Could this be due to the large number of neurons used in the GLM model? When analyzing area communication, methods (such as reduced ranking) use top components (with reduced dimension) to reduce over fitting.

The authors should at least try to use a different statistical method to validate major conclusions.

6. The authors used very complicated data analysis. It is important for the community to evaluate the code and also be benefited from the study. Although the authors planned to release the data and code, it is not possible to inspect the quality of deposited materials. A revised version ideally should have the deposition ready.

Minor:

1. The task should be described more clearly in the main text, especially for how tactile stimulus is applied. From the main text and figure 1a, it is unclear how left and right trials are determined.
2. Figure S3, each panel from one brain areas? Or some brain area but different session? Why pixels are much larger in some panel? Perhaps fewer neurons?
3. For the GCaMP line, expression is in excitatory neurons. For such wide expression, is there any epileptiform activity, like that observed in Ai93/Ai148?
4. Fig 6c, cell, unit, which one is simulation?

Reviewer #2 (Remarks to the Author):

Chia and colleagues leverage the ability to monitor cellular-resolution neuronal activity from eight cortical regions simultaneously in behaving mice to tackle the challenging problem of how learning shapes population dynamics in the brain. Combining this rich dataset with computational analyses, the authors show that functional connections carrying critical choice-related information are retained in the face of global sparsening induced by learning. The manuscript is well written, the figures are high quality and the methods are sound. However, as I detail below, the potential impact of the manuscript is somewhat dampened by the seemingly unwarranted focus on two regions (ALM and PPC). Moreover, there are potential analytical confounds that might impact the interpretation of some results, such as trial-type imbalance and the role of anticipatory licking. All of the issues can be addressed with expanded or modified analyses. With necessary revisions that will help solidify the author's claims, the manuscript will make important contributions to the field.

Major points

1. Despite the established roles of the ALM and PPC, recent work from many labs has highlighted the

importance of more distributed computations. Considering that the authors collected high-quality data on eight regions simultaneously, and that many of these regions appear to carry task-relevant information, the choice to focus on the connection between ALM and PPC in detriment of other areas appears somewhat arbitrary. It is possible that, as a consequence of this focus, the authors are missing important contributions from functional connectivity between other areas. Extending the current analyses (in particular those in Figs. 4 and 5) to explore the remaining area pairs will likely increase the impact of the manuscript.

2. Fig 2. An imbalance between the numbers of left and right trials could bring important confounds to these results. Naïve animals often display bias toward a particular side, which gradually diminishes over learning (e.g., there are many more left side datapoints in the naïve choice mode example in Fig. 2a). Thus, for the ALM–PPC activity space, the correlation across trial types may be pre-existing, but underestimated in the naïve case because there are very few points in one of the trial types, and only revealed for expert (unbiased) animals as both trial types become more evenly sampled. This confound could be controlled for by using trial subsampling. Analyses relying on the GLM (e.g., Fig 3) may also suffer from similar confounds, despite the regularization. It would be important to control for it in those cases as well.

3. Were animals penalized for licking preceding the response window? I could not find this in the methods section. If not, this is an important methodological confound, as the emerging choice signals could be explained by increased anticipatory licking. Since the authors have videos of the animals, this would be straightforwardly addressed by tracking tongue movements with deeplabcut, as in Fig. S1. This would have the added advantage of adding a useful predictor to the GLM, given the known role of the ALM in licking. More generally, it would be helpful to expand on methodological details about the task.

4. That learning-induced changes were not observed in the stimulus-mode across or within regions (Fig. 1f and 1g, respectively) contradicts recent findings that sensory-evoked signal emerge in frontal areas with learning (e.g., Le Merre et al. 2018, *Neuron*; Peters et al., 2022, *Cell Reports*). One possibility is that this is because the stimulus mode does not sufficiently capture the dynamics of stimulus selectivity. The mode axes are obtained by collapsing both temporal and trial-by-trial variability. A set of axes that is more sensitive to learning-induced changes could possibly be obtained by using approaches that take these factors into account, such as LDA, or by using the GLM weights instead of mean-subtracted trial averages. It would be important to show that the findings hold with more sensitive methods. It would also be important to discuss these findings in light of the papers mentioned above.

5. Asking whether particular ablations increase the likelihood of “choice switching” in Fig. 6 suggests a potentially interesting role of the tested connections. It would add a lot to the claim if the same principle could be directly applied to the neural data, e.g. using the GLM. One way this could be achieved is by selecting a set of predictors for ablation, and introducing a perturbation by scrambling the activity of the remaining predictors for a window during the delay period. The authors could then compare the likelihood of choice switching between the scrambled-with-ablation condition and the scrambled-without-ablation condition.

6. Although the findings in Fig. 6 are compelling, RNNs can be sensitive to hyperparameters such as g . It is important to show that the results are robust to changes in this and other hyperparameters.

7. Single-cell activity across learning stages is only shown for two example cells (Fig. 1c). Moreover, cell 1 appears to show emergence of stimulus selectivity, which is not observed at the population level (Figs. 1f and 1g). It would be very informative for the reader if average activity were shown for many more, if not all, recorded cells across the three learning stages, perhaps in a supplementary figure.

8. The discussion is a bit thin — it only summarizes the findings and mentions possible future directions. The section should be substantially expanded to place the findings in context of the wider present literature (e.g., see point 4).

Minor points

1. Fig. 2: The “Stimulus mode” and “Choice mode” labels appear to apply across all panels of the figure. It would be helpful for the reader if this were made more visually conspicuous.

2. Fig. 2c: Here (and in similar plots in Figs 3, 5, and S4) there is substantial overlap of the lines between the two areas of primary interest (ALM and PPC) and the lines between those areas and M1p. While the decision to preserve anatomical accuracy is understandable, some minor adjustments that reduce this overlap may be warranted here.

3. Fig. 5b: The change in enrichment index is not significant for the stimulus-encoding cells. However, there is a trend, which merits discussion along with the area-specific effects in Fig. S4a.

4. Lines 191–193: It is unclear what data substantiates this claim, given the sparsity of the increases in enrichment shown in Fig. 5d

5. Line 311: The legend should note from which regions the two example neurons were taken.

6. Line 492: “(...) a left or right water spout was made (...)” should be “(...) left and right water spouts were made (...)”, assuming both spouts were made available on each trial. If only one spout was made available, it is unclear how mice could make incorrect responses.

7. Lines 500–503: How (i.e., which and why) two out of the three first sessions were picked as naïve sessions should be noted, as well as how many intermediate sessions were selected from the middle of the interval.

8. Line 504: The authors should mention how many mice were excluded by this criterion.

9. Line 578: A minor caveat of the orthogonalization order (i.e., choice mode then stimulus mode) is that distances are better preserved in the choice mode than in the stimulus mode, because any interaction between the modes is retained in the choice mode and removed from the stimulus mode. It would be interesting to see whether the negative results for the stimulus mode hold even in the non-orthogonalized case.

10. I could not find methodological details on how neurons were registered across days. It would be important to add

Reviewer #1

Xin et al. aimed to study communications among neuron ensembles across multiple brain areas during short-term memory. They used 2P-RAM to chronically image populations of neurons in 8 brain areas (ALM, M1a, M1p, M2, S1fl, vS1, RSC and PPC) during the learning process. To analyze the stimulus and choice information during learning, the researchers projected neural activity onto two orthogonal axes that maximized the stimulus and choice selectivity during sample or pre-action epoch, respectively. They found stimulus selectivity remained constant after learning while choice selectivity increased across the dorsal cortex. The correlations between regions along the stimulus axis are relatively low compared the correlations along the choice axis. But the correlations along both axes increase with learning. The main finding of this paper is the cortical functional coupling generally reduced over learning, and this is caused by selective retention of choice-related coupling and elimination of choice-irrelevant coupling. Using a GLM model the authors simulated the choice activity of ALM with or without considering functional coupling of PPC-ALM, RSC-ALM and vS1-ALM. They found that only ablation of PPC-ALM coupling caused a significant reduction of choice related activity in ALM. In addition, after learning, PPC-ALM communication is critical for the robustness of choice-related attractor dynamics.

In summary, the authors used state-of-art meso-2P imaging to track the same population of neurons distributed in multiple regions during learning of a 2AFC task. The authors then used a set of encoding/network simulation methods to analyze the functional coupling during learning and provide some important insights about how long-range functional coupling potentially contributes to neural coding and network robustness. As large scale recording becomes routine and rodents are trained to perform more and more complex tasks, the methods/message conveyed by this manuscript is timely and important. Nevertheless, I have multiple questions/concerns listed below.

We are grateful to the reviewer for providing valuable and insightful feedback on our manuscript. Their supportive and constructive comments have been instrumental in improving our work. In the revised manuscript, we have conducted additional experiments to suppress PPC activity and carried out further analyses. We believe that these amendments have significantly strengthened our manuscript. To facilitate easy identification, we have highlighted the modifications to the original manuscript in yellow.

Major:

1. It is well known that functional connectivity might reflect common inputs, for example, from more stereotypic movements after learning. I agree that it is experimentally challenging to specifically manipulate functional coupling. Indeed, most optogenetic perturbations have not achieved cellular recapitulation of neural dynamics. However, coarse manipulation often leads to specific effects as behavior/neural dynamics is typically low dimensional. The authors should at least try to perturb PPC and investigate whether this manipulation affects ALM activity as simulated.

We agree with the reviewer that perturbation of PPC would further strengthen our finding that the PPC-ALM communication is important for the delayed-response task. In the revised manuscript, we performed two complementary experiments to suppress the PPC activity. First, we optogenetically silenced PPC in transgenic mice expressing Channelrhodopsin-2 (ChR2) in GABAergic neurons (VGAT-ChR2-EYFP), and evaluated resulting changes in their task performance. While this approach does not allow us to simultaneously image neural activity in ALM, we can temporally restrict PPC inactivation to the delay epoch. With PPC inhibition, the correct rate significantly reduced (**Figure 7a and b**), suggesting that PPC activity during the delay is critical for the task. In contrast, vS1 inactivation did not have any effect on the correct rate (**Figure 7b**), highlighting the specificity of PPC's contribution.

A previous study reported that, unlike our finding, PPC was dispensable during the delay period of a similar tactile task (Guo et al., 2014). Because training durations were much longer in Guo et al. (2014), we hypothesized that PPC becomes gradually less important as mice were trained longer. To test this, we kept training mice for additional 18-50 sessions and found that photoinhibition of PPC exerted little influence on the task performance of over-trained mice (**Figure 7b**). Such disengagement of the cortex in over-trained mice has been reported in other domains of learning (Hwang et al., 2021).

Second, we used designer receptors exclusively activated by designer drugs (DREADDs) by virally expressing hm4D(Gi)-mCherry in PPC of both hemispheres. We inactivated PPC while recording neural activity in ALM with two-photon calcium imaging. While this approach lacked temporal control of activity suppression, it enables us to simultaneously image ALM activity. We again found that the task performance was sensitive to PPC suppression (**Figure 7c, d, S10a and b**). Furthermore, with PPC inactivation, ALM neurons showed less choice selectivity (**Figure 7e**), which was measured by the

difference between right and left choice-related activity and served as an indicator of the task performance ($R^2 = 0.22$, $P < 0.01$, 32 sessions from 6 mice). Taken together, these results, described in line 220-248, page 7, support our conclusion that PPC is functionally relevant and PPC modulates ALM activity during the delayed-response task.

2. Related to the above, does PPC project to ALM?

According to the Allen brain mouse connectivity atlas and previous studies (Zhang et al., 2016; Zingg et al., 2014), PPC does not densely project to ALM. However, because PPC influences ALM activity during the delayed-response task (major point #1), it is likely that the activity of PPC was indirectly propagated to ALM through other brain structures. In contrast, while PPC heavily projects to M2 directly (Lyamzin and Benucci, 2019), we did not observe any strengthening of their interactions during learning. Thus, structural analysis of neural circuits does not fully capture learning-related dynamics in inter-areal communications. We now describe this important point in Discussion (line 306-318, page 9).

3. Related to 1. The enrichment index and choice activity increased from PPC->ALM. Could this be due to more stereotypic movements. Fig S1 only shows that movements on average did not increase.

We thank the reviewer for raising this important point. With DeepLabCut (Mathis et al., 2018), we analyzed right forelimb movements during the stimulus and pre-action epoch (starting 1 s prior to the go cue for 1 s). We focused on the right forelimb because our camera fully captured its movement while the view of the left forelimb was frequently blocked. As the reviewer correctly points out, right forelimb movements became more stereotyped across learning, as indicated by their increased trial-by-trial correlations (**Figure S2a-c**). However, changes in movement stereotypy were not related to changes in enrichment index for PPC-ALM (**Figure S7a**). Likewise, changes in movement stereotypy were unrelated to changes in coordination of choice activity between PPC and ALM (**Figure S2d**). These results indicate that our findings were unlikely to be explained by increased movement stereotypy. In the revised manuscript, we added these results in line 103-105, page 4 and line 175-177, page 5.

4. The authors observed that activity remained encoded choice better than activity eliminated. Could this be due to lasso regularization as this naturally removes components with small contribution?

We thank the reviewer for this insightful comment. In the original submission, we used the elastic net regularization with $\alpha = 0.95$ (lasso: 0.95 and ridge: 0.05). As the reviewer correctly points out, this tends to disregard GLM predictors with small contributions. In the revised manuscript, we trained GLMs using $\alpha = 0.5$ (lasso: 0.5 and ridge: 0.5) and $\alpha = 0.05$ (lasso: 0.05 and ridge: 0.95) to examine if our results were sensitive to these α values. Overall, while lower α values resulted in more retained cell-coupling across learning, our results remained essentially the same; reconstructed activity projected to the choice axis with retained cell-couplings was larger than reconstructed activity with eliminated cell-couplings (**Figure S6a and b**). We now describe this in line 159-160, page 5.

5. For the encoding model, the explained variance is only about 10%. A naïve question is: what's the meaning of analyzing different components of this 10% variance.

With training, task variables explain larger percentage of explained variance (compared with early training). I am not sure whether this is obvious in Fig. 3b. Could this be due to the large number of neurons used in the GLM model? When analyzing area communication, methods (such as reduced ranking) use top components (with reduced dimension) to reduce over fitting.

The authors should at least try to use a different statistical method to validate major conclusions.

We thank the reviewer for these important comments. In response to the first point, we looked into the distributions of pseudo-E.V. at each stage of learning and found that they were skewed and ranged up to ~0.6 (**Figure S4b**). These distributions were generally similar to those reported by other studies (e.g. (Steinmetz et al., 2019)). For each neuron, we statistically determined whether it encoded task variables or whether it received functional inputs from other neurons.

As the reviewer suggests, it is possible that contributions of task variables in GLM were masked by the large number of cell-coupling predictors. In addition, because the significant cell-coupling predictors decreased across learning (sparsening of functional connectivity), apparent contributions of the task variable predictors could increase. To address this, we performed area under the ROC

(receiver operating characteristic) curve (AUC) analysis and determined decoding accuracy of stimulus and choice in each neuron (**Figure 1e and f**). We also looked at GLM weights for the task variable predictors (**Figure S4c**). In general, decoding accuracy increased for both stimulus and choice across learning.

We further analyzed intra- and inter-regional communications with three independent methods. Firstly, we analyzed functional connectivity by computing pair-wise correlations of spontaneous activity during inter-trial intervals. Secondly, we measured pair-wise correlations of pre-action activity. These analyses revealed that learning led to sparser functional connectivity in the mouse dorsal cortex (**Figure S5b-h**). Thirdly, we performed reduced rank regression between population activity of source and target regions. In the revised manuscript, we now show that prediction performance from ALM to PPC and from PPC to ALM increased across learning (**Figure S8a-c**). These results were consistent with those obtained with the enrichment index analysis. We describe these results in line 72-77, page 3, line 137-141, page 4-5 and line 201-203, page 6 in the revised manuscript.

6. The authors used very complicated data analysis. It is important for the community to evaluate the code and also be benefited from the study. Although the authors planned to release the data and code, it is not possible to inspect the quality of deposited materials. A revised version ideally should have the deposition ready.

Under the sections of "Data availability" and "Code availability" in the revised manuscript, we provide links to GitHub and Zenodo where our analysis code and dataset are deposited, respectively.

Minor:

1. The task should be described more clearly in the main text, especially for how tactile stimulus is applied. From the main text and figure 1a, it is unclear how left and right trials are determined.

We apologize for the lack of detailed descriptions of the task. The tactile stimulus was presented with brass rods located on both sides of the snout, which swept the whiskers at ~20 Hz for 1 second. Left and right trials were randomly interleaved in each session. In the revised manuscript, we provide more information about the task in line 53-57, page 2.

2. Figure S3, each panel from one brain areas? Or some brain area but different session? Why pixels are much larger in some panel? Perhaps fewer neurons?

Each panel is an example session from an example mouse containing all commonly identified neurons across learning from all regions concatenated. The different pixel size is due to different number of neurons (larger pixels indicating fewer cells). We added this information in the legend of the revised manuscript (**Figure S5a**).

3. For the GCaMP line, expression is in excitatory neurons. For such wide expression, is there any epileptiform activity, like that observed in Ai93/Ai148?

We never observed any signs of epileptiform activity in this transgenic mouse line. We now describe this in the revised manuscript along with a citation of a previous study (Steinmetz et al., 2017) in line 71-72, page 3.

4. Fig 6c, cell, unit, which one is simulation?

"Cell" means neuron in the mouse cortex, while "unit" means RNN unit. We made them more explicit in the revised manuscript by referring them to as "Cell" and "RNN unit" in **Figure 8c**.

References

Guo, Z.V., Li, N., Huber, D., Ophir, E., Gutnisky, D., Ting, J.T., Feng, G., and Svoboda, K. (2014). Flow of cortical activity underlying a tactile decision in mice. *Neuron* 81, 179-194. 10.1016/j.neuron.2013.10.020.

Hwang, E.J., Dahlen, J.E., Mukundan, M., and Komiyama, T. (2021). Disengagement of Motor Cortex during Long-Term Learning Tracks the Performance Level of Learned Movements. *J Neurosci* 41, 7029-7047. 10.1523/JNEUROSCI.3049-20.2021.

Lyamzin, D., and Benucci, A. (2019). The mouse posterior parietal cortex: Anatomy and functions. *Neurosci Res* 140, 14-22. 10.1016/j.neures.2018.10.008.

Mathis, A., Mamidanna, P., Cury, K.M., Abe, T., Murthy, V.N., Mathis, M.W., and Bethge, M. (2018). DeepLabCut: markerless pose estimation of user-defined body parts with deep learning. *Nat Neurosci* 21, 1281-1289. 10.1038/s41593-018-0209-y.

Steinmetz, N.A., Buettner, C., Lecoq, J., Lee, C.R., Peters, A.J., Jacobs, E.A.K., Coen, P., Ollerenshaw, D.R., Valley, M.T., de Vries, S.E.J., et al. (2017). Aberrant Cortical Activity in Multiple GCaMP6-Expressing Transgenic Mouse Lines. *eNeuro* 4. 10.1523/ENEURO.0207-17.2017.

Steinmetz, N.A., Zatzka-Haas, P., Carandini, M., and Harris, K.D. (2019). Distributed coding of choice, action and engagement across the mouse brain. *Nature* 576, 266-273. 10.1038/s41586-019-1787-x.

Zhang, S., Xu, M., Chang, W.C., Ma, C., Hoang Do, J.P., Jeong, D., Lei, T., Fan, J.L., and Dan, Y. (2016). Organization of long-range inputs and outputs of frontal cortex for top-down control. *Nat Neurosci* 19, 1733-1742. 10.1038/nn.4417.

Zingg, B., Hintiryan, H., Gou, L., Song, M.Y., Bay, M., Bienkowski, M.S., Foster, N.N., Yamashita, S., Bowman, I., Toga, A.W., and Dong, H.W. (2014). Neural networks of the mouse neocortex. *Cell* 156, 1096-1111. 10.1016/j.cell.2014.02.023.

Reviewer #2

Chia and colleagues leverage the ability to monitor cellular-resolution neuronal activity from eight cortical regions simultaneously in behaving mice to tackle the challenging problem of how learning shapes population dynamics in the brain. Combining this rich dataset with computational analyses, the authors show that functional connections carrying critical choice-related information are retained in the face of global sparsening induced by learning. The manuscript is well written, the figures are high quality and the methods are sound. However, as I detail below, the potential impact of the manuscript is somewhat dampened by the seemingly unwarranted focus on two regions (ALM and PPC). Moreover, there are potential analytical confounds that might impact the interpretation of some results, such as trial-type imbalance and the role of anticipatory licking. All of the issues can be addressed with expanded or modified analyses. With necessary revisions that will help solidify the author's claims, the manuscript will make important contributions to the field.

We sincerely appreciate the reviewer's valuable feedback on our manuscript. As detailed below, we expanded our investigation to other cortical regions and carefully examined potential confounding variables. We believe that these amendments significantly improved our manuscript. For easy identification, we have highlighted the modifications to the original manuscript in yellow.

Major points

1. Despite the established roles of the ALM and PPC, recent work from many labs has highlighted the importance of more distributed computations. Considering that the authors collected high-quality data on eight regions simultaneously, and that many of these regions appear to carry task-relevant information, the choice to focus on the connection between ALM and PPC in detriment of other areas appears somewhat arbitrary. It is possible that, as a consequence of this focus, the authors are missing important contributions from functional connectivity between other areas. Extending the current analyses (in particular those in Figs. 4 and 5) to explore the remaining area pairs will likely increase the impact of the manuscript.

We agree that the main strength of our study is its unbiased data-driven approach that might potentially uncover previously under-explored cortical regions/connections. Our focus in the original manuscript was on the mutual interactions between PPC and ALM because the results were consistent across different analyses. We noted, however, that there was also learning-dependent strengthening in PPC-RSC interactions (originally in **Figure 5e-g, S5a and b**, now **Figure 6e-g and S9a-d**). To the best of our knowledge, this pathway had not been explored before in the context of short-term memory and our finding is potentially novel.

We believe that it is also informative to identify inter-regional communications that were not subject to changes during learning. In this regard, even though PPC projects heavily to M2, we did not find strong modulations in enrichment index between PPC and M2 (**Figure 6d**).

For the original **Figure 4a-d** (now **Figure 5a-d**), because we restricted our analysis to the cells that were commonly identified across the three learning stages, the limited number of cell-coupling terms between pairs of regions prevented us from determining region-pair-specific activity_{retained} and activity_{eliminated}. This is now mentioned in the revised manuscript (line 162-164, page 5).

In the revised manuscript, we performed PPC suppression experiments and found that PPC was important during the delayed-response task (**Figure 7a-d**). Thus, PPC may serve as an important hub to route choice information to other brain regions via PPC-ALM and PPC-RSC, but not PPC-M2 pathways. We now explain these findings in Discussion (line 299-318, page 9).

2. Fig 2. An imbalance between the numbers of left and right trials could bring important confounds to these results. Naïve animals often display bias toward a particular side, which gradually diminishes over learning (e.g., there are many more left side datapoints in the naïve choice mode example in Fig. 2a). Thus, for the ALM-PPC activity space, the correlation across trial types may be pre-existing, but underestimated in the naïve case because there are very few points in one of the trial types, and only revealed for expert (unbiased) animals as both trial types become more evenly sampled. This confound could be controlled for by using trial subsampling. Analyses relying on the GLM (e.g., Fig 3) may also suffer from similar confounds, despite the regularization. It would be important to control for it in those cases as well.

We thank the reviewer for raising these important points. As the reviewer correctly points out, the trial type imbalance could bias our results for the original **Figure 2a-c**, especially for naive mice. In the

revised manuscript, we subsampled trials to match the number of left and right trial types before computing the trial-by-trial correlations along the stimulus and choice axis. The result remained essentially the same (**Figure S3a and b**), lending support to our conclusion of enhanced intra- and inter-regional coordination across learning (now in **Figure 3a-c**).

For the original **Figure 3d-g**, fitting GLM over multiple times after subsampling trials to match the number of left and right trial types would be computationally expensive and time-consuming. Thus, we instead used additional two metrics to measure functional connectivity between neurons: (1) pairwise correlations of spontaneous activity during ITIs, which was insensitive to the trial type number mismatch and (2) pairwise trial-by-trial correlations of pre-action neural activity, which could be subject to a bias due to the trial type number mismatch. In the revised manuscript, we first show that these metrics display high similarities to the cell-coupling extracted from GLM before balancing the trial type number (**Figure S5b-e**, $P < 0.001$ for the three learning stages, Jaccard similarity). We also ensured that the correlation coefficients were not confounded by activity levels (**Figure S5f**). Then, we subsampled trials to match the trial type number and computed pairwise trial-by-trial correlations of pre-action neural activity. The sparsening of functional connectivity was still evident in both metrics (**Figure S5c and g**) and convergence index derived from this analysis (pairwise trial-by-trial correlations of pre-action neural activity after matching the trial type number, **Figure S5g**) was positively correlated with convergence index obtained with GLM (**Figure S5h**). Thus, gradual sparsening of functional connectivity across learning was not explained by the trial type number mismatch.

We now also demonstrate that increases in enrichment index were also not affected by the trial-type number mismatch (**Figure S7b**). These results are now described in line 105-106, page 4, line 137-141, page 4-5 and line 175-177, page 5.

3. Were animals penalized for licking preceding the response window? I could not find this in the methods section. If not, this is an important methodological confound, as the emerging choice signals could be explained by increased anticipatory licking. Since the authors have videos of the animals, this would be straightforwardly addressed by tracking tongue movements with deeplabcut, as in Fig. S1. This would have the added advantage of adding a useful predictor to the GLM, given the known role of the ALM in licking. More generally, it would be helpful to expand on methodological details about the task.

We again thank the reviewer for pointing out this potential confound. We confirmed that mice rarely showed anticipatory licking. In the delayed-response task, the left and right water spouts were not accessible during the delay period; both water spouts moved closer to mice upon the go cue. The first lick to either the left or right water spout during the response period was registered as their choice. We quantified anticipatory licking during the delay period by randomly sampling five-minute videos from seven mice at the naive and expert stage (total of 14 videos), which included an average of 20 trials per video. We detected zero anticipatory licking in four mice. In the other three mice, we detected only eight lick bouts out of ~120 trials. We believe such a low rate of anticipatory licking (less than 3%, 8 lick bouts in 280 trials) would not have a significant impact on the overall choice activity. We now describe this in the revised manuscript in line 86-89, page 3.

4. That learning-induced changes were not observed in the stimulus-mode across or within regions (Fig. 1f and 1g, respectively) contradicts recent findings that sensory-evoked signal emerge in frontal areas with learning (e.g., Le Merre et al. 2018, Neuron; Peters et al., 2022, Cell Reports). One possibility is that this is because the stimulus mode does not sufficiently capture the dynamics of stimulus selectivity. The mode axes are obtained by collapsing both temporal and trial-by-trial variability. A set of axes that is more sensitive to learning-induced changes could possibly be obtained by using approaches that take these factors into account, such as LDA, or by using the GLM weights instead of mean-subtracted trial averages. It would be important to show that the findings hold with more sensitive methods. It would also be important to discuss these findings in light of the papers mentioned above.

We truly thank the reviewer for this valuable feedback. Despite many experimental differences between these studies and ours, we agree that identifying potential consistency/inconsistency was important. To understand our data better in the context of these studies, we carried out additional analyses based on reviewer's suggestions. Firstly, we computed decoding accuracy for the stimulus location or choice in single cells or populations of cells projected to the stimulus or choice axis. With these more sensitive measurements taking temporal and trial-by-trial variability into account, we now find that decoding accuracy for these task variables increased in regions such as M2 (**Figure 1e, f, 2a and b**), which is consistent with these previous studies. GLM weights also generally showed similar trends (**Figure S4c**).

Because these measurements are more sensitive, we replaced the results in the original manuscript with these new findings, which are now explained in line 72-77, page 3 and line 81-84, page 3.

5. Asking whether particular ablations increase the likelihood of “choice switching” in Fig. 6 suggests a potentially interesting role of the tested connections. It would add a lot to the claim if the same principle could be directly applied to the neural data, e.g. using the GLM. One way this could be achieved is by selecting a set of predictors for ablation, and introducing a perturbation by scrambling the activity of the remaining predictors for a window during the delay period. The authors could then compare the likelihood of choice switching between the scrambled-with-ablation condition and the scrambled-without-ablation condition.

We thank the reviewer for this thought-provoking suggestion. As far as we understand the comment, scrambling the neural activity during the delay period (similar to the perturbation in RNNs) would modulate the neural activity during the pre-action epoch (between -1 and 0 s relative to the go cue). This is true for RNNs where the activity at later time points (i.e. pre-action epoch) depends on the activity at earlier time points (i.e. during perturbation). However, the neural activity of the mouse brain reconstructed with GLM cannot be manipulated in the same way; scrambling the neural activity during the early phase of the delay period to mimic perturbation does not alter the neural activity during the pre-action period. Thus, we replaced the neural activity during the delay with the neural activity scrambled during ITIs instead, and now demonstrate that the resulting decrease in the choice activity was similar to that derived from the original removal of cell-coupling terms (**Figure S9c and d**). We now describe this result in line 216-217, page 6.

6. Although the findings in Fig. 6 are compelling, RNNs can be sensitive to hyperparameters such as g . It is important to show that the results are robust to changes in this and other hyperparameters.

Following the suggestion, we explored the hyperparameter space of g along with different distractor amplitudes. We found that the results remained generally the same; PPC-ALM activity, compared to vS1-ALM activity, rendered RNNs more robust to perturbation (**Figure S11d**). We explain this result in line 266-267, page 8 of the revised manuscript.

7. Single-cell activity across learning stages is only shown for two example cells (Fig. 1c). Moreover, cell 1 appears to show emergence of stimulus selectivity, which is not observed at the population level (Figs. 1f and 1g). It would be very informative for the reader if average activity were shown for many more, if not all, recorded cells across the three learning stages, perhaps in a supplementary figure.

In **Figure S1** of the revised manuscript, we now show four example neurons from each cortical region across the three learning stages.

8. The discussion is a bit thin — it only summarizes the findings and mentions possible future directions. The section should be substantially expanded to place the findings in context of the wider present literature (e.g., see point 4).

We agree with the reviewer that the original Discussion was not comprehensive. In the revised manuscript, we expanded the Discussion section by detailing potential importance of other cortico-cortical communications during learning of the delayed-response task and changes in neural representation of stimulus information across learning, in the context of previous findings (line 299-324, page 9).

Minor points

1. Fig. 2: The “Stimulus mode” and “Choice mode” labels appear to apply across all panels of the figure. It would be helpful for the reader if this were made more visually conspicuous.

We thank the reviewer for the suggestion. We modified **Figure 3** by adding a line between the stimulus and choice mode.

2. Fig. 2c: Here (and in similar plots in Figs 3, 5, and S4) there is substantial overlap of the lines between the two areas of primary interest (ALM and PPC) and the lines between those areas and M1p. While the decision to preserve anatomical accuracy is understandable, some minor adjustments that reduce this overlap may be warranted here.

In the revised manuscript, we modified these figure panels (**Figure 3c, 4g, 6d, S3b and S7d**) by slightly shifting the ALM node so that individual lines are more visible.

3. Fig. 5b: The change in enrichment index is not significant for the stimulus-encoding cells. However, there is a trend, which merits discussion along with the area-specific effects in Fig. S4a.

In the revised manuscript, we now describe this in detail in line 177-178, page 5 and line 186-190, page 6. Please also refer to the major point #4 above.

4. Lines 191–193: It is unclear what data substantiates this claim, given the sparsity of the increases in enrichment shown in Fig. 5d

We thank the reviewer for catching this. In the original manuscript, we wrote “*The increase in the inter-regional enrichment indices was more prominent than the intra-regional enrichment indices, highlighting the recruitment of long-range functional connectivity during learning.*”. In the original **Figure 5d** (now **Figure 6d**), we showed matrices describing changes in enrichment index across learning. For the choice code, most of the significant increases in enrichment index were detected in off-diagonal entries (except the PPC-PPC intra-regional enrichment index), suggesting that the increase in inter-regional enrichment index was more prominent than the increase in intra-regional enrichment index. However, we agree with the reviewer that this claim was not generally substantiated by our data and decided to remove this sentence in the revised manuscript.

5. Line 311: The legend should note from which regions the two example neurons were taken.

In **Figure 1d**, we now describe where these example neurons were selected from (cell 1: ALM; cell 2: PPC). To address the major point #7 above, we now show more examples in **Figure S1**.

6. Line 492: “(...) a left or right water spout was made (...)” should be “(...) left and right water spouts were made (...)”, assuming both spouts were made available on each trial. If only one spout was made available, it is unclear how mice could make incorrect responses.

We apologize for the lack of clarity. In the delayed-response task, both left and right water spouts moved together at the beginning of the go cue and made available to mice so that they could lick either the left or right water spout. The first lick event during the response period of each trial was registered as an response. We now describe this in line 53-57, page 2.

7. Lines 500–503: How (i.e., which and why) two out of the three first sessions were picked as naive sessions should be noted, as well as how many intermediate sessions were selected from the middle of the interval.

We now describe this in Methods (line 403-409, page 34) as “*For each mouse, naive, intermediate, and expert sessions (two sessions for each learning stage) were determined. The naive sessions were defined as the first and second sessions by default. However, the first session for one mouse was excluded because there were fewer than five responses (correct or incorrect) for the left trial type. The first sessions for two mice were also excluded due to technical issues encountered during the recording of their behavior while they were properly trained. The expert sessions were defined as the first two sessions where the correct rate consistently reached ~70-75%. The intermediate sessions were selected from midpoints of the naive and expert sessions.*”

8. Line 504: The authors should mention how many mice were excluded by this criterion.

Two mice were excluded by this criterion, which is now described in Methods (line 410-411, page 34).

9. Line 578: A minor caveat of the orthogonalization order (i.e., choice mode then stimulus mode) is that distances are better preserved in the choice mode than in the stimulus mode, because any interaction between the modes is retained in the choice mode and removed from the stimulus mode. It would be interesting to see whether the negative results for the stimulus mode hold even in the non-orthogonalized case.

We thank the reviewer for this comment. It is true that the order of orthogonalization could alter the results. Below, we now show a non-orthogonalized case for the population activity projected to the stimulus axis. Overall, the results are very similar to **Figure 2a**, demonstrating that decoding accuracy (related to the major point #4) for stimulus location increased across learning in several cortical regions. We now mention this in the revised manuscript (line 81-84, page 3).

10. I could not find methodological details on how neurons were registered across days. It would be important to add

We used ROIMatchPub (<https://github.com/ransona/ROIMatchPub>) to register the same neurons across sessions. In the revised manuscript, we described this in Methods (line 484-489, page 36).

REVIEWERS' COMMENTS

Reviewer #2 (Remarks to the Author):

The authors did a good job addressing my concerns, and as a result I found the manuscript to be much improved. A very minor note for the final version is that I did not follow the scrambling procedures addressing my point 5 (Fig S9c,d), and could not find a description in the revised text. I'd recommend adding a sentence or two in the Methods section. I congratulate the authors on an interesting story.

Reviewer #2

The authors did a good job addressing my concerns, and as a result I found the manuscript to be much improved. A very minor note for the final version is that I did not follow the scrambling procedures addressing my point 5 (Fig S9c,d), and could not find a description in the revised text. I'd recommend adding a sentence or two in the Methods section. I congratulate the authors on an interesting story.

We thank the reviewer for their support for the publication of our manuscript. For **Figure S9c and d**, we included a relevant description in the main text (line 216-217, page 6) and a paragraph in the Methods section (line 689-692, page 19).